**ARTICLES**
# In vivo lensless microscopy via a phase mask generating diffraction patterns with high-contrast contours

Jesse K. Adams [1,2,6], Dong Yan [1,2,6], Jimin Wu [3,6], Vivek Boominathan [2,6], Sibo Gao[2,4], Alex V. Rodriguez[2], Soonyoung Kim [2], Jennifer Carns [3], Rebecca Richards-Kortum [2,3], Caleb Kemere [2,3,4], Ashok Veeraraghavan [1,2,5] ✉ and Jacob T. Robinson [1,2,3,4] ✉

The simple and compact optics of lensless microscopes and the associated computational algorithms allow for large fields of view and the refocusing of the captured images. However, existing lensless techniques cannot accurately reconstruct the typical low-contrast images of optically dense biological tissue. Here we show that lensless imaging of tissue in vivo can be achieved via an optical phase mask designed to create a point spread function consisting of high-contrast contours with a broad spectrum of spatial frequencies. We built a prototype lensless microscope incorporating the 'contour' phase mask and used it to image calcium dynamics in the cortex of live mice (over a field of view of about 16 mm²) and in freely moving *Hydra vulgaris*, as well as microvasculature in the oral mucosa of volunteers. The low cost, small form factor and computational refocusing capability of in vivo lensless microscopy may open it up to clinical uses, especially for imaging difficult-to-reach areas of the body.

Computational imaging algorithms, co-designed with optics and sensors, improve the design options for imaging systems[1–3]. For example, computational imaging techniques have already made a substantial impact on microscopy, for applications such as super-resolution[4,5], phase imaging[6] and three-dimensional (3D) imaging[7,8]. Taken to an extreme, computation can completely replace the traditional lens(es) of an optical system[9], or be combined with light-modulating masks[10,11]. Unlike lens-based imaging systems where the goal is to project a reproduction of a (potentially magnified) scene onto an image sensor, a lensless imaging system seeks to produce an invertible transfer function between the incident light field and the sensor measurements. These measurements often may not resemble a traditional image[12–18], but contain sufficient information for a computational algorithm to reconstruct an image.

By removing lenses from the image capture process, systems can achieve substantial improvements in the field of view (FOV), perform 3D capture and refocusing, capture light with high efficiency and be miniaturized into small form factors[10–12,19].

Despite the transformative potential of lensless microscopy, there are currently no demonstrations of in vivo biological imaging due to the fact that biological scenes are dense, dim and low-contrast, which can often result in noisy reconstructed images. Indeed, previous lensless microscopy demonstrations have relied on sparse, bright or high-contrast samples where strong regularization and/or deblurring can be used to reconstruct estimates of the original image[10,19,20].

To address the challenge of biological imaging, we designed a phase mask especially suited for the low-contrast challenges expected in biological imaging. The key feature of this phase mask is that it achieves a high-contrast and spatially localized sparse point spread function (PSF) that captures textural frequencies common in natural and biological samples (Fig. 1d). To create this PSF, we generate Perlin noise and apply Canny edge detection[3]. Figure 1b shows the simulated modulation transfer function (MTF) of our contour-based PSF compared to other PSFs used in lensless imaging systems. The relatively flat MTF spectrum indicates that information from most spatial frequencies is well-preserved by the optical transfer function, leading to improved image reconstruction for dense, low-contrast samples like biological tissue. We chose to create this PSF using a phase mask (Supplementary Fig. 2) because phase masks are capable of producing a wide variety of PSFs[3,21–26] and allow higher light throughput compared with amplitude masks (which block some of the incoming light)[10]. The phase mask was designed to produce the target contour-based PSF using a near-field phase retrieval algorithm[3,27]. We refer to this prototype device (Fig. 1a) as 'Bio-FlatScope' because it achieves a similar flat form factor as the previously reported 'FlatScope'[10], yet the contour phase mask allows us to perform accurate lensless imaging of biological tissue in vivo.

## Results

To evaluate the performance of Bio-FlatScope, we constructed prototypes using Sony IMX178 monochromatic 6 MP imaging sensors with 2.4 μm pixels. Phase masks were fabricated using a two-photon lithography system (Nanoscribe, Photonic Professional GT) for ease of prototyping (see Methods). The contour-based phase masks were designed with a minimum feature width of 6–12 μm.

**Spatial resolution and 3D imaging in fixed biological samples.** Using USAF resolution targets, we found that reconstructed images taken by the Bio-FlatScope achieve <9 μm lateral resolution.

[1]Applied Physics Program, Rice University, Houston, TX, USA. [2]Department of Electrical and Computer Engineering, Rice University, Houston, TX, USA. [3]Department of Bioengineering, Rice University, Houston, TX, USA. [4]Department of Neuroscience, Baylor College of Medicine, Houston, TX, USA. [5]Department of Computer Science, Rice University, Houston, TX, USA. [6]These authors contributed equally: Jesse K. Adams, Dong Yan, Jimin Wu, Vivek Boominathan. ✉e-mail: vashok@rice.edu; jtrobinson@rice.edu

We measured this resolution by capturing images of a negative 1951 USAF Resolution Target (Edmund Optics 59-204) with an added fluorescent background. Multiple images were captured at 15 ms each and averaged to remove noise. Excitation light was provided in a near-epifluorescence (near-epi) configuration with a fibre/micro-prism (Supplementary Fig. 3). Images were captured at a distance of ~4 mm. In Fig. 1d, we show that we can achieve <9 μm resolution.

The main advantage of the Bio-FlatScope is its ability to accurately reconstruct dense, dim and low-contrast biological samples; thus, we compare the performance of Bio-FlatScope to other lensless imaging techniques for imaging a fluorescent biological sample. Specifically, we compare reconstructed images of *Convallaria majalis* (lily of the valley) stained with green fluorescent protein (GFP). We notice that for the high-contrast USAF test targets, both the FlatScope and Bio-FlatScope show clear image reconstructions; however, for the *Convallaria*, only the Bio-FlatScope shows good correspondence to ground truth images captured with a ×4 microscope objective (Nikon Fluor) (Fig. 1d). These data are based on multiple images of the *Convallaria* slice (at ~3.5 mm from the device) captured at 200 ms each in the near-epi configuration and averaged for noise removal. Note in the *Convallaria* zoom-ins that for the lensless approaches tested, only the Bio-FlatScope can resolve some of the large circular plant cells with sizes of around 10 μm. We also confirm that Bio-FlatScope can resolve images in fixed neural tissue using ex vivo mouse brain slices (deep cortex and hippocampus) expressing the fluorescent protein GCaMP6f (see Methods). Here we capture 20 images at a distance of ~5.3 mm using transmissive illumination with 1 s exposures, averaged to remove noise. Comparisons between Bio-FlatScope images and ground-truth microscopy show clear delineation between cortex layer 2/3, corpus callosum and hippocampus (CA1) (Supplementary Fig. 5). We can also identify clusters of neurons in the cortex as well as the pyramidal layer of CA1 (see far-right column in Supplementary Fig. 5).

An additional advantage for lensless imaging is the ability to obtain 3D information with a single image capture, due to the fact that the PSFs change as a function of depth[3,11]. To characterize the 3D imaging ability, we prepared a 3D test sample by suspending 10 μm fluorescent beads in a ~3 × 3 × 0.5 mm³ clear phantom. For ground truth, we captured the 3D fluorescent phantom using a scanning confocal microscope (Fig. 2a). Bio-FlatScope images were captured at a distance of ~2.8 mm and reconstructions were performed using PSFs from 2.8 mm to 3.3 mm. We find excellent agreement between the Bio-FlatScope reconstruction (Fig. 2a) and the confocal data over a depth range of 500 μm. Empirically, we found that the full width at half maximum (FWHM) of the axial spread of the 10 μm beads is approximately 50 μm in the Bio-FlatScope reconstructions (Fig. 2b), which indicates a 50 μm axial resolution. Next, to evaluate the performance of Bio-FlatScope in scattering media, we prepared a 3D sample using the same protocol as the clear phantom, but with the addition of 1 μm non-fluorescent polymer beads to mimic the scattering property of tissue. The scattering strength is controlled by the density of the polymer beads. For our experiments, we chose the scattering coefficient of ~1 mm⁻¹ to be similar to mouse brain tissue[28,29]. When we reconstructed images from this scattering phantom, we found that the FWHM of the axial spread of the 10 μm beads is approximately 80 μm in the Bio-FlatScope reconstructions (Fig. 2e), which indicates an 80 μm axial resolution in scattering media.

To better demonstrate that Bio-FlatScope is capable of single-shot 3D reconstruction of biological samples, we imaged a fixed *Hydra vulgaris* expressing GFP in the ectoderm with Bio-FlatScope and compared the reconstructions to a confocal stack as ground truth. *Hydra* samples were captured by Bio-FlatScope immediately after being prepared (see Methods). Ground truth images of *Hydra* samples were then taken using a confocal microscope immediately after Bio-FlatScope imaging. The reconstruction of Bio-FlatScope clearly indicates that the depth information of the *Hydra* closely corresponds to the ground truth image (Fig. 2g). Given this ability, we also used the Bio-Flatscope to perform live imaging of calcium activity in freely moving, GCaMP7b-expressing *Hydra* (Supplementary Fig. 4).

**In vivo epifluorescence calcium imaging in mice.** We also found that the Bio-FlatScope is capable of imaging calcium dynamics in mouse cortex, which is commonly used for neuroscience experiments and preclinical testing[30–33]. This type of in vivo imaging is particularly difficult for lensless imaging and epifluorescence imaging because tissue scattering causes blurring and loss of contrast. Haemodynamic changes were not considered in our reconstruction and data analysis.

We hypothesized that the optimized contour PSF with high-contrast features will provide good protection against noise amplification in these low-contrast scenes and allow Bio-FlatScope to perform reconstructions through scattering media similar to epifluorescence imaging (Supplementary Fig. 6).

To test our hypothesis, mice expressing GCaMP6f in the motor cortex (see Methods) were head-fixed and placed on a freely moving treadmill to allow for locomotion during video capture (Fig. 3a). We recorded calcium imaging movies over an FOV of ~16 mm² of the entire cranial window (Supplementary Fig. 8) using a 475 nm excitation light source (see Methods). A cropped region near the injection site is shown in Fig. 3c. Compared with previously reported miniature microscopes with cellular resolution[34], we achieve more than a 30× increase in the FOV (see Supplementary Table 1). We also measured the velocity of the treadmill during imaging sessions and synchronized the image captures with this velocity data. During the recording sessions, we applied a brushing tactile stimulation at 30 s intervals to encourage locomotion and neural response in the motor cortex. Correlation between locomotion (as performed by the mouse here on a treadmill) and neuronal response in the motor cortex has been established in previous reports[35–37] and is confirmed in our epifluorescence imaging. In the regions of high activity in the motor cortex, we observed that peak fluorescence signals during locomotion are on average ~3× greater than during periods of no motion (root mean square), and were able to resolve blood vessels as small as ~10 μm in diameter. To compare our Bio-FlatScope imaging to conventional epifluorescence, we replaced the Bio-FlatScope with a ×4 objective lens and fluorescence microscope. We performed epifluorescence imaging under the same conditions within 30 min of our Bio-FlatScope recordings (see Methods) and found very similar spatiotemporal dynamics in the two datasets (Fig. 3, and Supplementary Figs. 9 and 10). We chose the ×4 objective lens to accommodate the FOV captured by Bio-FlatScope (while maintaining high resolution) and registered both datasets to accommodate for movement[38,39].

To compare Bio-FlatScope data with epifluorescence data, we selected 4 s windows time-aligned to the response to stimulation (reflected as a large change in ambulatory activity of the mouse) over a 2 min period (Fig. 3d,e, shown in green). Ambulatory activity was determined by synchronizing the wheel velocity with the imaging data for both Bio-FlatScope and epifluorescence (shown in Fig. 3d,e for a single region of interest (ROI); Supplementary Video 2 shows video for multiple regions). We validated the similarity of the windowed ΔF/F responses with correlation, finding a median correlation coefficient of 0.852, with interquartile range (IQR) of 0.131 (Fig. 3f, left side). We also compared data from time windows of little to no locomotion, where velocity was <1 cm s⁻¹ (windows shown in Supplementary Fig. 9), with epifluorescence-captured motion resulting in a median correlation coefficient of −0.375 and an IQR of 0.396 (Fig. 3f, right side). As expected, data from regions of motion/stimuli highly correlate when comparing Bio-FlatScope to epifluorescence data (coefficients range from 0.65 to 0.97, P < 0.001), while regions of little to no motion show more random correlation (coefficients ranging from −0.52 to 0.31, two values of P < 0.001

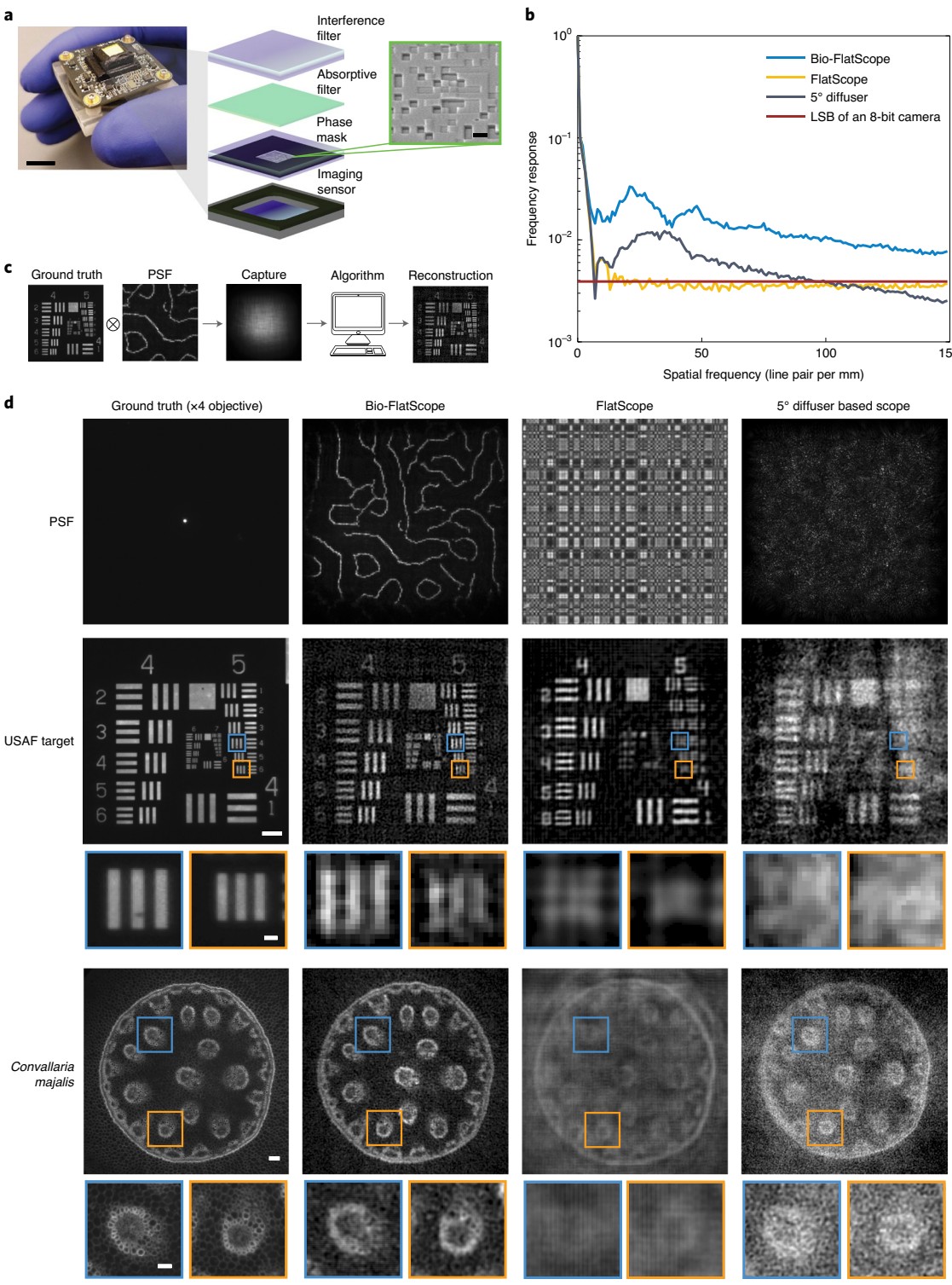

**Fig. 1 | Bio-FlatScope for in vivo fluorescence imaging. a**, Photo of a Bio-FlatScope prototype including the off-the-shelf board level camera. Scale bar, 10 mm. Zoom-in shows components that filter excitation light and apply the phase mask transfer function to the incident light. Also shown is a scanned electron micrograph of a portion of the fabricated phase mask. Scale bar, 4 μm. **b**, MTF comparison among PSF designs used by lensless imaging systems[10,11] shows that the contour PSF used for the Bio-FlatScope contains more high-frequency components, which contributes to improved performance. LSB, least significant bit. **c**, Flowchart showing the imaging procedure of Bio-FlatScope. **d**, High spatial resolution images of a USAF test target and fixed biological samples. Top: PSF captures of different lensless imaging systems. Middle and bottom: ground truth of high-contrast USAF target and *Convallaria majalis*, respectively, captured by an epifluorescence microscope and reconstructed by 3 different lensless imaging systems. While FlatScope and Bio-FlatScope both show good performance for the high-contrast USAF target, only the Bio-FlatScope performs well on the low-contrast, dense *Convallaria* sample. Zoom-ins below middle row show comparisons of group 5, elements of 4 and 6, and zoom-ins below bottom row show comparisons of highlighted structures . Scale bars: USAF test target, 100 μm; group 5 zoom-ins, 10 μm; *C. majalis* test target, 100 μm; *C. majalis* zoom-ins, 50 μm.

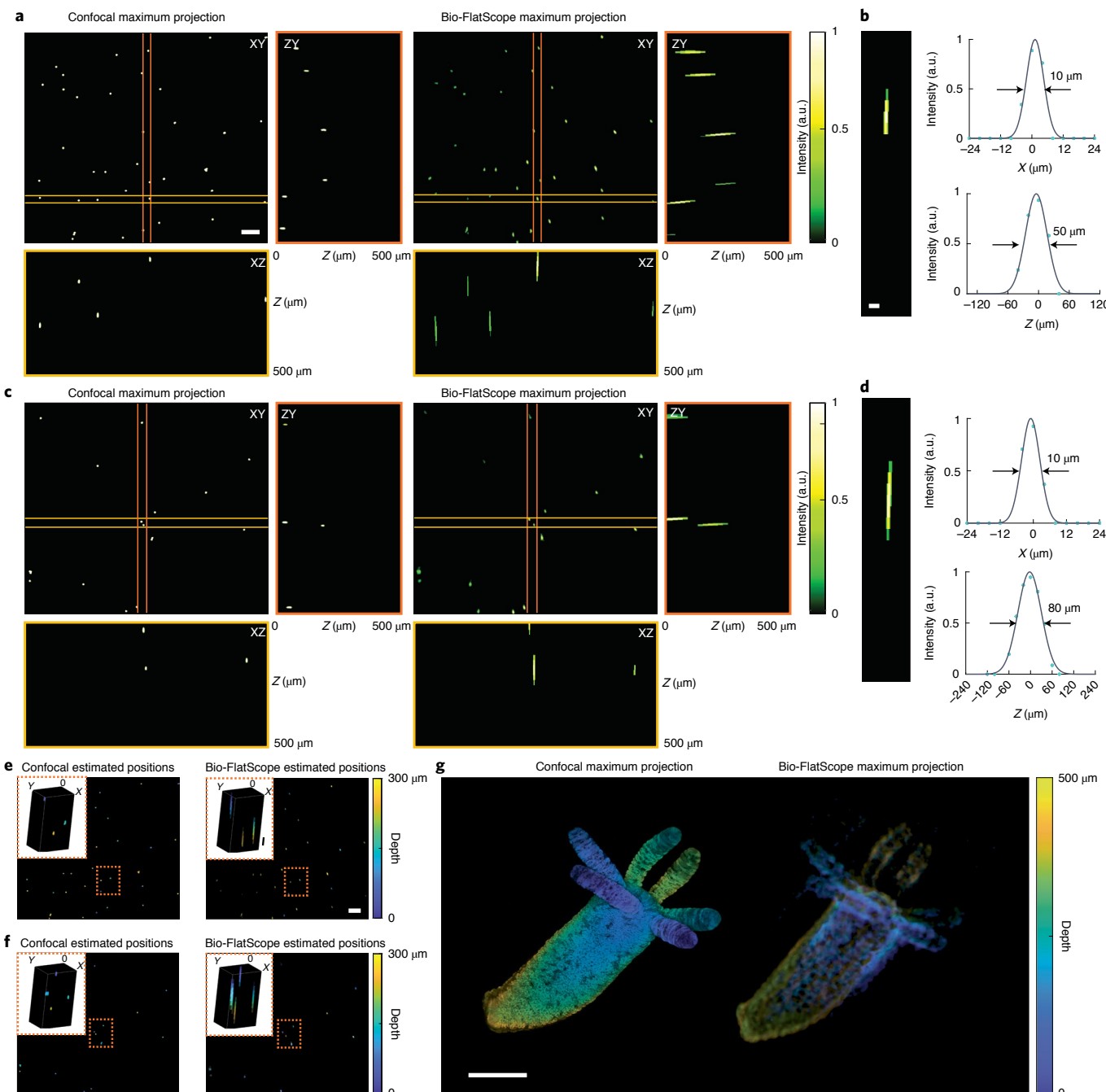

**Fig. 2 | 3D volume reconstruction of fixed fluorescent samples. a**, Bio-FlatScope reconstruction in non-scattering media as a maximum intensity projection along the Z axis as well as a ZY slice and an XZ slice, compared with the ground truth captured by confocal microscopy (×10 objective). Scale bar, 100 μm. **b**, Axial (XZ) profile (left), and corresponding X (top right) and Z (bottom right) profiles of a reconstructed bead in non-scattering media, showing lateral and axial resolution, respectively. Scale bar, 20 μm. **c**, Bio-FlatScope reconstruction in scattering media as a maximum intensity projection along the Z axis as well as a ZY slice and an XZ slice, compared with the ground truth captured by confocal microscopy (×10 objective). Scale bar the same as in **a**. **d**, Axial (XZ) profile (left), and corresponding X (top right) and Z (bottom right) profiles of a reconstructed bead in scattering media, showing lateral and axial resolution, respectively. Scale bar the same as in **b**. **e**, Depth-color-coded maximum projection of Bio-FlatScope reconstruction in non-scattering media, compared with the ground truth captured by confocal microscopy (×10 objective). Scale bar, 100 μm. Zoom-in shows a 3D volume of a selected area. Scale bar, 20 μm (Z axis). **f**, Depth-color-coded maximum projection of Bio-FlatScope reconstruction in scattering media, compared with the ground truth captured by confocal microscopy (×10 objective). Scale bar the same as in **e**. **g**, Depth-color-coded maximum projection of Bio-FlatScope reconstruction of a fixed *Hydra* sample, compared with the ground truth captured by confocal microscopy (×10 objective). Scale bar, 500 μm.

and six values of $P > 0.05$), confirming that using Bio-FlatScope, we can capture behaviourally relevant calcium dynamics that are comparable to conventional dynamics capture with epifluorescence microscopes.

In addition, when plotting the time of the calcium peaks over the FOV, we find that when the animal initiates movement, the majority of early peaks appear in the bottom and right of the FOV, and the majority of late peaks appear in the top and left (Fig. 3g). This

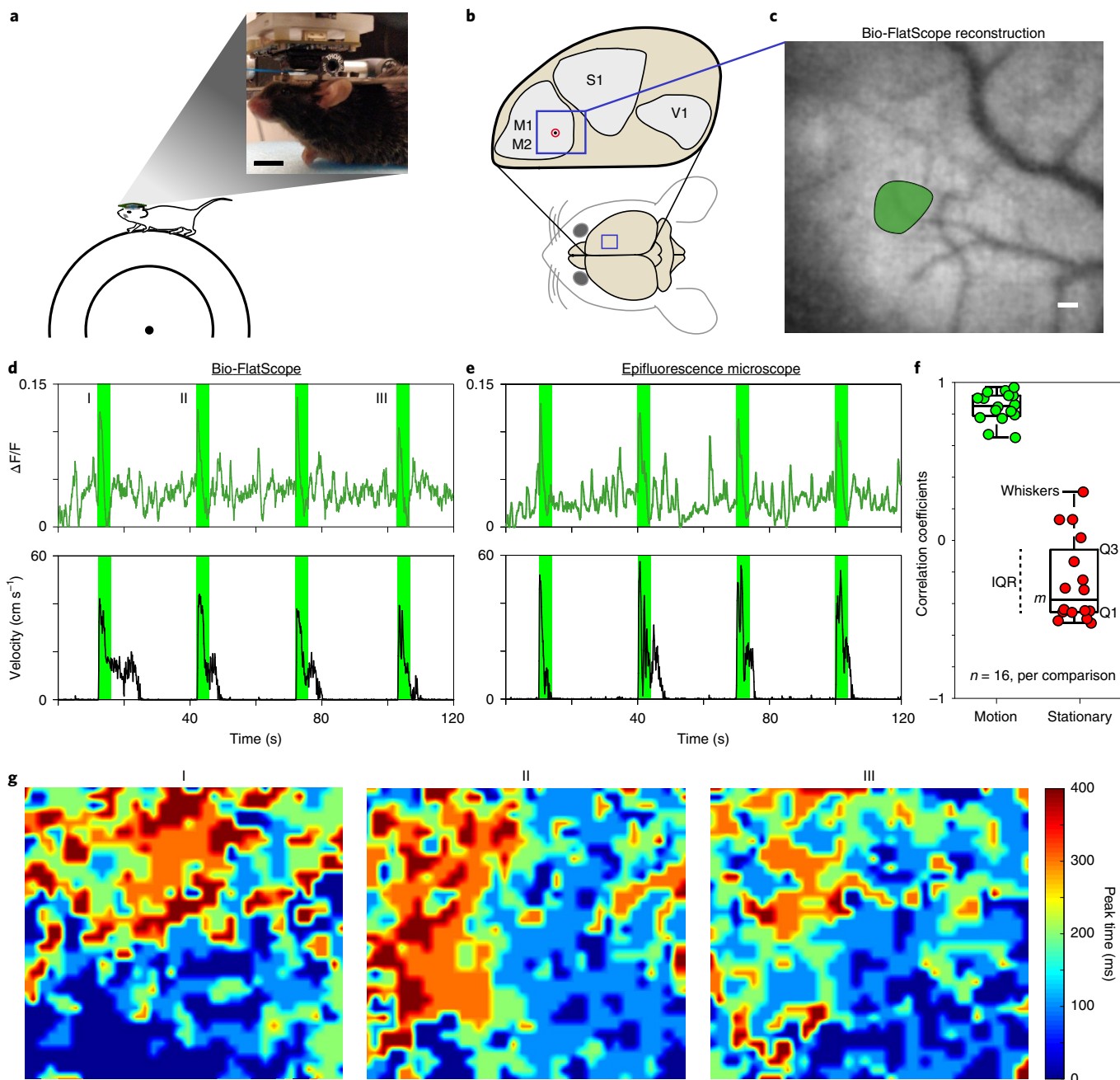

**Fig. 3 | Comparison of Bio-FlatScope to epifluorescence microscopy during stimulus-evoked Ca$^{2+}$ responses in motor cortex. a**, Experimental setup of Bio-FlatScope for imaging a head-fixed mouse on a freely moving treadmill. Zoom-in shows a photo of a mouse on the treadmill, with Bio-FlatScope in place. Scale bar, 10 mm. **b**, Target region in the mouse brain. The blue square region indicates the approximate location of the cropped FOV (in the motor cortex). M1: primary motor cortex; M2: supplementary motor area; S1: primary somatosensory cortex; V1: primary visual cortex. The circle with the black dot indicates the approximate region of injection of GCaMP6f. **c**, Bio-FlatScope reconstruction with a single ROI of high activity marked. Scale bar, 100 μm. **d,e**, ΔF/F trace and treadmill velocity for Bio-FlatScope (**d**) and epifluorescence microscope (**e**) during the recording session. The rising edges of the 4 s windows in green correspond to the application of tactile stimuli. **f**, Boxplot showing correlation coefficients comparing stimulus-evoked Ca$^{2+}$ responses from Bio-FlatScope and epifluorescence (shown in green), and comparison of periods of little or no motion with Bio-FlatScope to stimulus-evoked Ca$^{2+}$ responses with epifluorescence (shown in red) for the ROI. The median ($m = 0.852$ for motion-to-motion, $m = -0.375$ for stationary-to-motion), IQR Q1/Q3 (25th and 75th percentiles) and whiskers (corresponding to minima and maxima, no outliers) are shown for $n = 16$ cross-correlation points over a total of 8 time windows from 2 independent experiments. **g**, Heatmaps for Bio-FlatScope reconstructions showing spatiotemporal Ca$^{2+}$ dynamics time-aligned with stimuli (at I, II and III). Colourmap shows the time at which pixels have their peak response for ΔF/F during a 400 ms period. These data show that when initiating movement, neural activity consistently begins in the lower right and propagates to the upper left.

pattern is also observed with epifluorescence imaging, confirming that Bio-FlatScope measures similar spatiotemporal Ca$^{2+}$ dynamics (images were processed for visualization, see Supplementary Fig. 10

for epifluorescence data). The recorded video was captured at 10 Hz (50 ms exposures) and shows a period of 400 ms following the tactile stimulation.

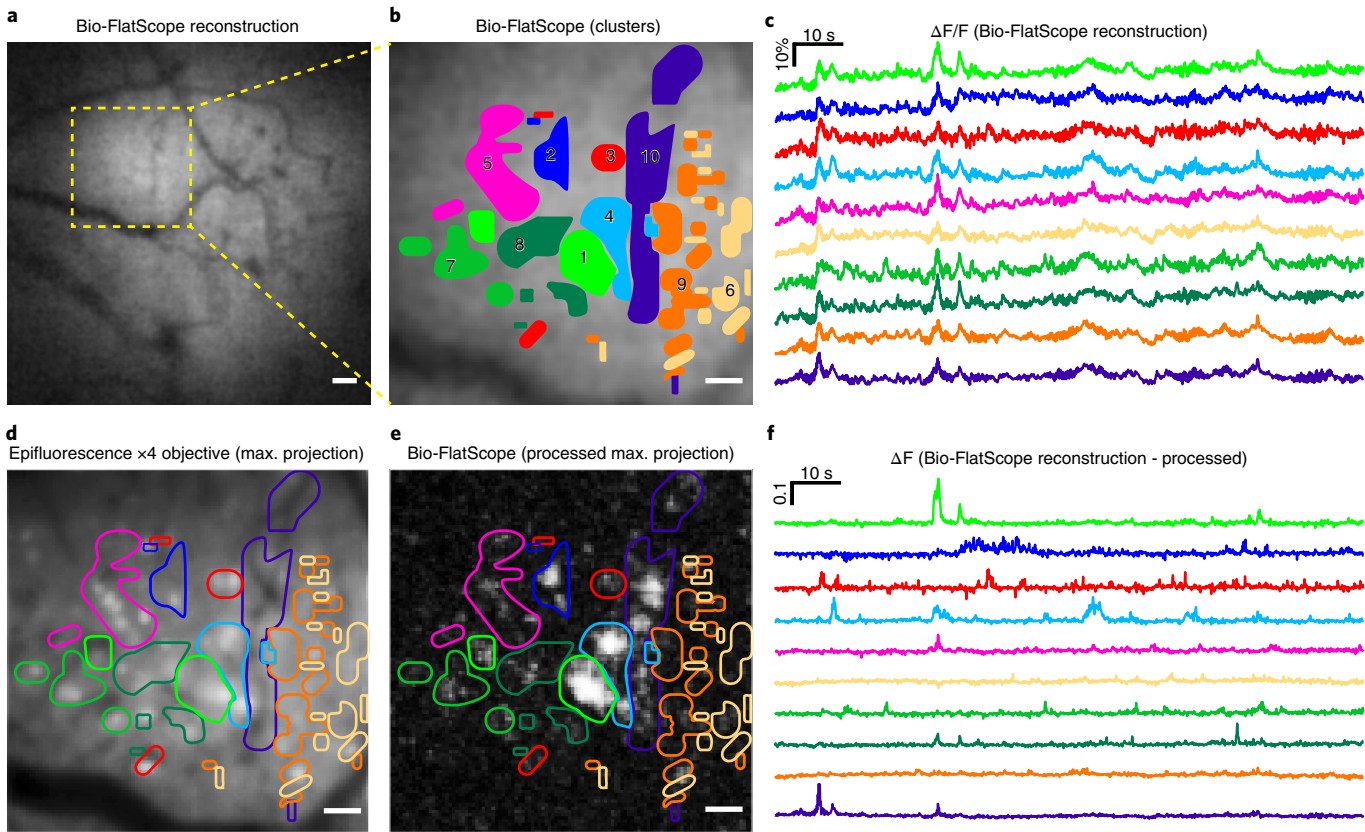

**Fig. 4 | Extracting Ca²⁺ signals from Bio-FlatScope reconstructions. a**, Cropped region of Bio-FlatScope reconstruction of a single frame with a high-activity region shown by a dashed box. Scale bar, 100 μm. **b**, Zoom-in on the region of high activity, with overlay of clusters determined through post processing using RPCA and *K*-means. Scale bar, 50 μm. **c**, ΔF/F traces from Bio-FlatScope reconstruction video during 2 min recording corresponding to the clusters. **d**, Maximum projection from epifluorescence recording of the same high-activity region, with overlay of clusters determined from Bio-FlatScope data. Scale bar, 50 μm. **e**, Maximum intensity projection of Bio-FlatScope reconstruction data processed using RPCA, with overlay of clusters. Scale bar, 50 μm. **f**, ΔF traces from Bio-FlatScope reconstructions after post processing using RPCA.

We can also extract unique calcium activity from small ROIs. The calcium activity in these ROIs (see Supplementary Table 3 for areas) shows unique temporal dynamics, suggesting that calcium dynamics might be recovered from Bio-FlatScope captures with high spatial resolution. To capture this data, we rely on the fact that in the time domain, some Ca²⁺ activity in individual or small groups of cells appears as outliers compared with the background activity. Robust principal component analysis (RPCA) excels at determining such sparse outliers[40,41], thus we used the RPCA algorithm to identify these regions of high calcium activity. We then used *K*-means to identify which high-activity regions show correlated activity. For comparison, this process was also performed on epifluorescence data (Supplementary Fig. 18) with the clusters found using the Bio-FlatScope data, noting that epifluorescence images were captured during the same session but at a separate time from the Bio-Flatscope images. When we compared the ROIs determined through RPCA and *K*-means from Bio-FlatScope, we found close correspondence to bright individual neurons observed from epifluorescence microscopy (Fig. 4d).

**Imaging of the human oral mucosa.** In addition to preclinical imaging, we found that the Bio-FlatScope can image human microvasculature in vivo, which is a clinically relevant biomarker for diseases like sepsis[42,43], oral neoplasia[44–46] and oesophageal neoplasia[47–50]. To test our ability to image microvasculature, we imaged oral mucosa in healthy human volunteers. The Bio-FlatScope prototype was placed a few millimetres away from the inner

surface of the lower lip to image the microvasculature in the labial mucosa using transillumination with a green light-emitting diode (LED) coupled to a liquid light guide in contact with the outer surface of the lower lip (Fig. 5). We can see in Fig. 5a,b that Bio-FlatScope reconstructed the morphological characteristics of microvasculature in labial mucosa, with contrast comparable to a ×4 objective and with the ability to resolve microvessels less than 20 μm in diameter. The measurements in Fig. 5d,e were based on images captured from 10 different nearby regions. Importantly, we found that individual differences in microvascular density measured with a ×4 objective lens correlate with those measured using the Bio-FlatScope. For this comparison, we captured ×4 and Bio-FlatScope images from approximately the same region of the oral mucosa; however, because there was no way to accurately register images between recording modalities, all measurements represent slightly different regions of tissue. Changes in microvessel density are important in a number of clinical scenarios: angiogenesis is a hallmark of cancer[51]; microvessel density is correlated with prognosis for a number of cancers[52,53]; and sub 20 μm vessel density has been proposed as a biomarker of sepsis[42,43].

The ability to quantify microvascular density combined with the ability to measure vessel morphology suggests that our lensless imaging technology could be used for clinical microvascular imaging, with key advantages associated with lensless imaging. Specifically, our small form factor is useful for accessing difficult-to-reach areas of the body and is advantageous for low-resource settings where high-resolution in vivo imaging

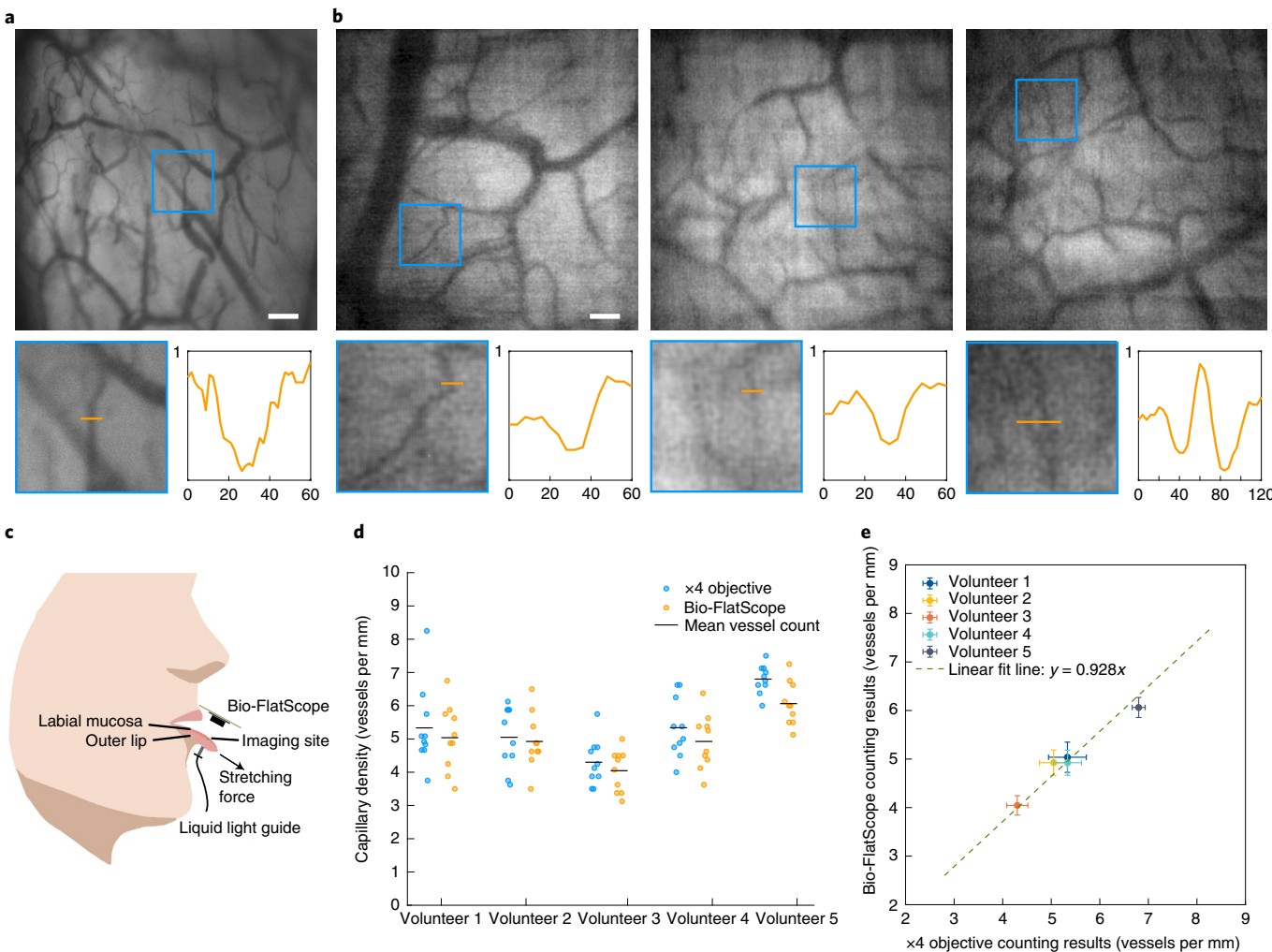

**Fig. 5 | Human oral mucosa imaging. a**, Images acquired using a ×4 objective. Scale bar, 200 μm. **b**, Reconstructed images from a Bio-FlatScope prototype. In **a** and **b**, each blue square indicates a subregion that contains ~15-μm-diameter vessels. Scale bar, 200 μm. Left zoom-ins: subregions. Right zoom-ins: pixel intensity along the cross-section (shown as orange bars) of the vessels. Length of orange bars in zoom-ins, 60 μm (120 μm in the 4th zoom-in). X axis, distance (in μm) from the left origin of the orange bar; Y axis, pixel intensity in normalized arbitrary units, shared by all 4 plots. **c**, Experimental setup. The transmissive illumination is delivered by a liquid light guide connected to a green LED light source. **d**, Vessel counting results of each volunteer. Each outlined circle corresponds to a different region on the lip. Less translucency indicates overlapping dots. There are 10 circles in each column. **e**, Relationship between ×4 objective and Bio-FlatScope counting results. Each circle is the average of 10 different lip regions as shown in **d**. Note that we imaged the same approximate region of the labial mucosa with both the ×4 objective and Bio-FlatScope, but were unable to capture images of the exact same area of tissue when switching between imaging modalities. Error bars, s.e.m.

techniques may be prohibitively expensive. This approach enables more patient-friendly form factors and larger fields of view compared with conventional microscopes. In addition, Bio-FlatScope has advantages over conventional microscopy as the ability to capture 3D information and to computationally refocus could enable rapid imaging over curved surfaces and provide additional diagnostic information.

## Discussion

Here we have demonstrated structural and functional imaging of biological samples using lensless microscopes with high spatial and temporal resolutions, large FOVs >16 mm², computational refocusing and single-capture 3D imaging over volumes of 4 mm³. The key innovation, a contour PSF, enables in vivo $Ca^{2+}$ imaging and human microvascular imaging with a lensless microscope.

Key next steps will be to improve the packaging and electronic components to make compact devices that could be used for preclinical studies of dynamic calcium activity, endoscopy and biomedical imaging of targets located at low depth. In the case of $Ca^{2+}$ imaging, we were able to extract localized $Ca^{2+}$ signals ranging in area from 0.001 mm² to 0.011 mm² using RPCA. These results serve as proof-of-principle that lensless microscopy provides a route towards high-resolution and large-FOV calcium imaging in a small lightweight form factor. By reducing (or rearranging) the electronic components around the imaging sensor, we foresee miniaturizing our device to the point that we can mount it atop a mouse or small animal to study calcium activity during unrestricted behaviour, similar to mini-scopes, but with a superior combination of spatial resolution and field of view.

While the resolution shown in fixed samples (Figs. 1 and 2) appears to meet the needs for microvascular imaging and $Ca^{2+}$ imaging, the in vivo resolution could become worse as the scenes are usually dense, dim, low-contrast and with increased complexity[10,11]. Although these challenges have already been greatly

overcome by Bio-FlatScope, there is still room for improvement. This improvement could be realized by optimizing the masks for closer scene-to-mask distances. These closer distances will not only improve spatial and axial resolution, but will also increase light collection efficiency and reduce the form factor, enabling easier integration. Additionally, integration of the device can allow for a smaller refractive index change between the sample and the device surface, which can cause a slight barrel effect at the edge of the FOV. As one brings the device closer to the sample, there is a corresponding decrease in the FOV until it matches the size of the sensor. To increase the effective FOV, future devices could be arrayed to maintain large scene coverage while maintaining a thin device profile.

The light-scattering tissue between the plane of interest and the imager also causes reduced performance in vivo. It is important to note that the blurring and loss of contrast resulting from scattering affects the properties of the scene but has no direct effect on the performance of our reconstruction algorithm because the reconstruction algorithm makes no assumptions about the medium between the source and the imager. Compared with epifluorescence microscopy, the only additional effect of scattering on lensless imaging is the fact that noise is amplified by the reconstruction algorithm, which is true for all lensless imaging reconstruction techniques, and mitigated by the contour PSF of Bio-FlatScope. It should be noted that as implemented here, the Bio-Flatscope does not overcome any of the scattering challenges that face traditional epifluorescence microscopy. Nevertheless, Bio-FlatScope does have the advantage of allowing us to computationally refocus after we capture the data. This computational refocusing allows us to form focused images of superficial regions typically within 200 μm of the surface of the tissue[54]. This ability is particularly useful for neural imaging, as this post-processing step can be used to account for axial position change of the brain surface and curvature of the brain. With the computational refocusing ability of Bio-FlatScope, the axial position change of the brain surface can be accounted for in computational post-processing by selecting PSFs at different depths during reconstruction. The key advantage of refocusing during post processing is that it does not require the additional hardware and real-time systems needed to perform the physical manipulation of the optics that is necessary for autofocusing during an experiment. An example of the utility of computational refocusing is shown in our imaging of human oral mucosa (Supplementary Fig. 14).

Other modifications to the design could further improve performance. Integration of the excitation light source (or multiple sources) can be introduced to improve the illumination coverage over the FOV. For this purpose, commercially available micro-LEDs or integrated waveguides can be used at little to no cost to the form factor. To improve on the low-light capabilities necessary for quality fluorescence microscopy, scientific complementary metal-oxide-semiconductor (sCMOS) imaging sensors or single photon avalanche diodes (SPADs) could be integrated to increase the signal-to-noise ratio, which is very important in lensless imaging. In addition to physical alterations of the lensless system, integration of new computational methods for imaging through scattering media might be used to extract additional 3D information for brain imaging. One important opportunity for this class of lensless in vivo imagers is the potential to perform large area, high-resolution imaging in freely moving animals. While our current prototype, which weighs ~5 g, is too heavy for a freely moving mouse, nearly all this weight comes from the printed circuit board and electronic components that support the CMOS image sensor (Supplementary Fig. 19). For other miniature head-mounted microscopes, the sensor printed circuit board (PCB) and electronics have already been made light enough to be head mounted to a freely moving mouse (Supplementary Table 1). Thus, using a similar design, we expect that future Bio-FlatScope will be compatible with freely moving mouse experiments. Additionally, because our system is designed to capture incoherent sources, other imaging modalities such as bright-field, darkfield and reflected-light microscopy are also possible, as we demonstrated in the case of microvascular imaging without the use of any contrast agents. These lensless microscopy modalities could be useful for endoscopy and point-of-care diagnostics.

Overall, technologies based on the Bio-FlatScope can be applied to a number of challenging microscopy problems where high temporal resolution, large FOV and small form factor are outside the capabilities of traditional lens-based microscopy.

## Methods

**Fabrication.** Bio-FlatScope prototypes were constructed using an off-the-shelf board level camera (Imaging Source, DMM 37UX178-ML) with a monochrome Sony CMOS imaging sensor (IMX178LLJ, 6.3 MP, 2.4 μm pixels). The glass cover on the CMOS image sensor (~1.5 mm thick, including air gap) prevents ultra-close sensor-to-mask distances. In our case, we operated with a ~4 mm working distance and a ~3.5 mm sensor-to-mask distance, so we did not need to remove this cover when building our prototypes. For closer sensor-to-scene prototypes one may choose sensors with removable glass cover (for example, Rasp Pi camera), or with a smaller gap between the glass cover and the active sensor surface.

The phase mask was fabricated using a 3D maskless two-photon photolithography system (Nanoscribe, Photonic Professional GT) with the high-resolution dip-in liquid lithography (DiLL) configuration, using a photoresist (Nanoscribe, IP-Dip) on a 700-μm-thick fused silica substrate. The laser power and writing speed may vary across systems and should be adjusted such that the pattern can be clearly observed in the real-time monitoring software (Nanowrite or AxioVision, with proper camera parameters adjustment). After the exposure, the sample was soaked in SU-8 developer for 20 min, and then soaked in isopropyl alcohol for 2 min. The fabricated phase mask has an overall height of 1 μm with a 200 nm height step, and a pixel size of 1 or 2 μm. The substrate was then cut down to more closely match the sensor size of the camera (and filters). The substrate was masked with an opaque material to provide an aperture containing only the phase elements. The substrate with phase mask was affixed atop the imaging sensor, followed by a hybrid filter, similar to Richard et al.[55,56], but using a commercially available absorptive filter (Kodak, Wratten 12) placed below an interference filter (Chroma, ET525/50m), both cut to the sensor size requirements (see Fig. 1a). One potential challenge for lensless fluorescence imaging is the fact that absorption filters designed to filter the excitation light can themselves produce autofluorescence. One approach to overcome this issue was demonstrated by Sasagawa et al.[57], which used a fibre-optic plate to help collimate the autofluorescence so that it would be better rejected by an interference filter. In our case, however, we found that increasing the working distance and separation between the absorption filter and the sensor for FlatScope[10] and Bio-FlatScope allowed us to achieve sufficient fluorescence contrast because it reduced the amount of excitation light incident on the filter, and improved the performance of the interference filter by constraining the angle of the incident light. A housing was 3D printed (MJP 2500) to hold the phase mask and filters atop the imaging sensor.

**Calibration.** The experimental PSF of the mask must be acquired through a one-time calibration process before capturing scenes. Because we cannot capture images of an ideal point source, we instead used a single 10 μm fluorescent microsphere (FluoSpheres yellow-green). The fluorescent bead was dropcast onto a microscope slide, then protected by a coverslip (Nexterion, 170 μm thick). Images of the PSF were captured for each depth (every 20 μm) over a distance range of 2–7 mm. This 'distance', which is mentioned throughout the text, is defined as the distance between the outermost surface of the Bio-Flatscope (the interference filter) and the imaging plane, which is similar to a microscope working distance. The bead was approximately aligned to the centre of the phase mask. Calibration images were averaged through multiple captures and background was subtracted to ensure the highest signal-to-noise ratio for the PSFs. The sensitivity of the system to depth requires calibration images of PSFs to be taken over a range of distances, which allows for refocusing and 3D reconstruction in post processing.

**Reconstruction.** To reconstruct images, we effectively solved the following minimization problem, adding Tikhonov regularization to the deconvolution to avoid noise amplification:

$$\widehat{\mathbf{i}} = \arg\min_{\mathbf{i}} ||\mathbf{b} - p * \mathbf{i}||_F^2 + \frac{\gamma}{2} ||\mathbf{i}||_2^2$$

where * denotes convolution, $\widehat{\mathbf{i}}$ is the estimate of the scene, $\mathbf{i}$ is the scene, $\mathbf{b}$ is the sensor measurement, $p$ is the PSF at the scene depth, $\gamma$ is the regularization weight, and $|| \cdot ||_F$ is the Frobenius norm. For reduced computational complexity, this minimization problem can be solved by closed form via Wiener deconvolution as:

$$\widehat{\mathbf{i}} = \mathcal{F}^{-1}\left( \frac{\mathcal{F}(p)^* \odot \mathcal{F}(\mathbf{b})}{|\mathcal{F}(p)|^2 + \gamma} \right)$$

where ⊙ denotes Hadamard product, $(\cdot)^*$ is the complex conjugate operator, $\mathcal{F}$ denotes a Fourier transform and $\mathcal{F}^{-1}$ denotes an inverse Fourier transform. The current runtime on an 8-core processor (AMD Ryzen 7 3700X, 3.59 GHz) is around 1.3 s for a 2D frame containing 3,072 × 2,048 pixels using MATLAB. Each frame was stored as a 16-bit TIFF file. This runtime can probably be improved through algorithm optimization.

Three-dimensional reconstruction from a single 2D capture is still an under-determined problem, and compressed sensing theory provides some guidance on samples that can be reliably reconstructed[58–60]. To reconstruct three-dimensionally, we solved the same minimization problem, but did so for the sum of depths of interest and with additional total variation (TV) regularization term $||\Psi(\mathbf{i}_d)||_1$ and L1 regularization term $||\mathbf{i}||_1$:

$$\hat{\mathbf{i}} = \arg\min_{\mathbf{i}\geq 0}\left\|\mathbf{b} - \sum_{d=1}^{D} p_d * \mathbf{i}_d\right\|_F^2 + \gamma_1\sum_{d=1}^{D}||\Psi(\mathbf{i}_d)||_1 + \gamma_2||\mathbf{i}||_1$$

where $D$ is the number of depths, $d$ is the distance from the imaging device and $\Psi$ denotes 2D spatial gradient. The TV and L1 regularization terms were added under the sparsity assumption. The 3D minimization problem was solved using alternating direction method of multipliers (ADMM)[61].

In our experiments, the plane for reconstruction was chosen by selecting the depth at which features of interest (for example, blood vessels) were reconstructed with the best sharpness in the region of interest.

**Excitation light.** Excitation light for Bio-FlatScope was achieved both transmissively and in near-epi configurations. Transmissive illumination was provided either using light through an epifluorescent microscope objective (Nikon Fluor ×4, NA 0.13 PhL DL), or using a near-collimated 470 nm LED (Thorlabs, M470L3) with an incorporated GFP excitation filter (Thorlabs, MF469-35). Near-epi illumination was provided using a fibre-coupled 475 nm LED (Prizmatix, UHP-T-475-SR) with an incorporated GFP excitation filter (Thorlabs, MF469-35). The light was coupled into a multimode fibre-optic cable (Thorlabs M72L01, 200 μm diameter core, NA 0.39; or Edmund Optics, 57-749, 400 μm diameter core, NA 0.22) with an aluminum-coated microprism (Edmund Optics, 0.18 mm 66-768 or 0.70 mm 66-773) adhered to the exposed tip of the fibre with optical epoxy (Norland, NOA72). The microprism was placed near the surface of the lensless imaging device. The multimode fibre directs the light from the blue excitation LED source, and the aluminum-coated microprism works as a mirror to redirect the light emitting from the fibre to the surface of the mouse brain. As a result, the final light spot is similar to the typical output pattern of a LED source. The response of our device to different illumination intensities is characterized in Supplementary Fig. 7.

The excitation light irradiation area of our fibre-optic cable and microprism can be increased by: (1) increasing the NA of the fibre-optic cable (limited by the maximum NA of commercially available optical fibres) and (2) increasing the distance between the microprism and the biological sample being illuminated. For the specific needs of our Bio-FlatScope prototype, the optimal excitation adjustment would have an area equal to the FOV of the prototype while maintaining sufficient illumination intensity.

**3D fluorescence measurements.** The 3D sample was prepared with 10 μm fluorescent beads ($3.6 \times 10^4$ beads ml$^{-1}$) in polydimethylsiloxane (PDMS) (Sylgard, Dow Corning; 10:1 elastomer:cross-linker weight ratio). Samples were cured at room temperature for a minimum of 24 h. The scattering phantom was prepared by suspending non-fluorescent polystyrene microspheres (Polybead Acrylate Microspheres, 1 μm diameter, $4.55 \times 10^{10}$ beads ml$^{-1}$) in PDMS (Sylgard, Dow Corning; 10:1 elastomer:cross-linker weight ratio). A concentration of $5.46 \times 10^9$ beads ml$^{-1}$ was used to achieve a scattering coefficient of ~1 mm$^{-1}$ to simulate the reported scattering coefficients in mouse brain tissue[28,29]. Microspheres (0.6 ml) were added to isopropyl alcohol (0.6 ml) and vortexed thoroughly for uniformity. The microsphere/alcohol mixture was added to the PDMS elastomer and mixed thoroughly, followed by cross-linker and again mixed thoroughly for uniformity. For the 3D scattering phantom, the samples were cured at room temperature for a minimum of 24 h, and the desired thickness of 500 μm was achieved by cutting a thick phantom. For the brain tissue phantom, the desired thickness of 140 μm was achieved by spin-coating the mixture onto a SiO$_2$ wafer, which was then heated at 37 °C for 30 min.

Ground-truth images of the 3D sample were captured with a depth range of 500 μm using a confocal microscope (Nikon Eclipse Ti/Nikon, CFI Plan Apo, ×10 objective). The confocal imaging required scanning and stitching to match the FOV of Bio-FlatScope; Z-axis measurements were captured every 2 μm. The excitation light for all fluorescence samples captured by Bio-FlatScope was provided by a 470 nm LED (Thorlabs M470L3) with an excitation filter (BrightLine Basic 469/35).

**Mouse brain slice.** All experiments were approved by the Rice University Institutional Animal Care and Use Committee (IACUC). Mice were killed 14 d post injection of AAV9-CamKII-GCaMP6f in the CA1 region of the hippocampus. From bregma, the injection site was +2 mm medial/lateral, −2 mm anterior/

posterior and −1.65 mm deep. The total injection amount was 0.5 μl at a rate of 0.05 μl min$^{-1}$. The brain expressing GCaMP6f was cryoprotected in 30% sucrose, embedded in optimal cutting temperature compound (Tissue-Tek), frozen and sliced into 50 μm slices using a cryostat (Leica). Slices were then rinsed in 1× phosphate-buffered saline (PBS), mounted using Vectashield H-1000 with DAPI (Vector Labs) and then sealed with coverslips.

***Hydra vulgaris.*** *Hydra* were cultured in *Hydra* media using the protocol adapted from the Steele Lab (UC Irvine) at 18 °C with 12:12 h light:dark cycle. Animals were fed freshly hatched *Artemia* every 2 d and cleaned after 4 h. Transgenic lines developed by embryo microinjections[62–68] expressing GFP in interstitial cell lineage (GFP, Neurons), GCaMP7b in ectodermal muscle cells (GCaMP7b, Ecto), and GFP in the ectoderm and RFP in the endoderm (Watermelon *Hydra*) were used. The transgenic line of *Hydra* expressing GFP under actin promoter was originally developed by the Steele Lab and selected for expression in neurons. The transgenic line of *Hydra* expressing GCaMP7b under EF1a promoter was developed by the Robinson Lab and the Juliano Lab (UC Davis) and selected for expression in ectodermal muscle cells. The transgenic line nGreen, kindly provided by Rob Steele, was generated by microinjecting embryos with a plasmid containing the *Hydra* Actin promoter driving GFP expression.

In fixed-sample imaging using Watermelon *Hydra*, the *Hydra* were treated with 4% formaldehyde diluted in *Hydra* media from a 16% formaldehyde solution (Thermo Fisher 28906) for 30 s to ensure that they stayed in the same position over time. Then *Hydra* were immediately transferred to a Petri dish with No. 1.5 glass coverslip bottom (MatTek P35G-1.5-14-C), and 1% agarose solution in *Hydra* media was poured on top to fix the *Hydra* in place. The *Hydra* was positioned so that the whole body was angled for more depth coverage. The agarose solution was cooled down to avoid damaging the *Hydra*. Samples were made fresh before imaging.

Both *Hydra* expressing GFP and GCaMP7b were imaged in *Hydra* media on Petri dishes with coverslip bottoms (MatTek P35G-1.5-14-C). Bio-FlatScope images were captured through the coverslip with excitation light provided by the fibre-optic cable/microprism combination. The GCaMP7b-expressing *Hydra* was provided with sufficient media to move around freely, in this case ~3–5 mm of depth within the Petri dish. Supplementary Video 1 shows reconstruction video from a single depth plane, hence showing the *Hydra* 'leaving the image and returning'.

**Illumination strategy for fluorescence imaging.** For fluorescence imaging, we illuminated the sample with a fibre-optic cable attached to an aluminum-coated microprism[29]. This illumination strategy is capable of fitting into the small space between the lensless imager and the biological sample. To prevent the excitation light from reaching the sensor and overwhelming the fluorescence signal, we constructed a hybrid excitation filter that combines an absorptive and interference filter[55] and attached it to the top of the phase mask. The function of the interference filter is to reject the majority of blue excitation light scattered and reflected from the sample to the sensor. The absorptive filter, of optical density 3 (OD3), works well to reject the remaining off-angle excitation light that does not get rejected well by the interference filter. The thickness between the image sensor and the surface of the device is ~5 mm.

**Animals for in vivo mouse brain imaging.** Wild-type C57BL/6 mice ($n=7$) from Charles River Laboratories were used for this study. Animals were housed with standard 12 h light/dark cycle with ad libitum food and water. All animals ($n=7$) were injected with the adeno-associated viral vector AAV9.CamKII.GCaMP6f. WPRE.SV40 (Penn Vector Core). A total of 7 mice were imaged in the experiments and reconstructions can be made with these images. We selected 2 images in the Results on the basis of the expression level, surgery performance and mouse activity (shown in Figs. 3 (and Supplementary Fig. 8) and 4 (and Supplementary Fig. 13)). All experimental procedures were approved by the IACUC at Rice University and followed the guidelines of the National Institutes of Health.

**Headpost implant and cranial-window design.** The headpost implant design was adapted from Ghanbari et al.[69], and consists of a custom-made titanium or stainless-steel head-plate, a 3D-printed frame, and three 0-80 screws to hold the frame to the head-plate. We fabricated the head-plate with titanium or stainless-steel plate (McMaster-Carr) using a waterjet system (OMax), and 3D printed the frame using a ProJet MJP 2500 (3D Systems). Our design files can be found online (https://github.com/ckemere/TreadmillTracker/tree/master/UMinnHeadposts). We assembled the headpost implant after tapping the 3D-printed frame with 0-80 tap and securing the head-plate over the frame with three screws. The entire headpost was then stored in 70% ethanol before surgery.

The cranial-window fabrication procedure was adapted from Goldey et al.[70]. Windows were made of 2 stacked round coverslips (Warner Instruments CS-3R, CS-4R, CS-5R) of different diameters. To fabricate the stacked windows, a 3 mm (or 4 mm) round coverslip was epoxied to a 4 mm (or 5 mm) round coverslip using an optical adhesive (Norland Products; for example, NOA 61, 71, 84) and cured using long-wavelength UV light. To accommodate the large 5 mm stacked window, we cut off the right side of the 3D-printed frame to allow for extra space for the C&B

Metabond to bind to the skull outside of the stacked cranial window. Fabricated stacked windows were stored in 70% ethanol before surgery.

**Surgical procedures.** For AAV9 injections, mice were anaesthetized with 1–2% isoflurane gas in oxygen and administered sustained release Buprenorphine SR-LAB (Zoofarm, 0.5 mg kg$^{-1}$). A small craniotomy was carefully drilled at the target location. A total of 0.5–1 µl of AAV9 virus was injected slowly at a rate of 0.07 µl min$^{-1}$ into each mouse with a Hamilton syringe paired with syringe pump controller (KD Scientific 78-0311). Specifically, mouse W1 was injected at −1.5 AP; +1.5 ML; −0.25 DV targeting the motor cortex; mouse W2 was injected at −1.67 AP; +1.1 ML; −0.25 DV targeting the motor cortex; mouse W3 was injected at −1.23 AP; +1.23 ML; −0.25 DV targeting the motor cortex. Following the injection, a small amount of bone wax was applied over the craniotomy, while taking care not to press down on the brain surface. The incisions were closed using a small drop of Vetbond (3M), and the mice were allowed to express for at least 4 weeks before headpost and window implantation.

For headpost and window implantation, mice were administered Buprenorphine SR-LAB (0.5 mg kg$^{-1}$) and Dexamethasone (2 mg kg$^{-1}$) 30 min before the craniotomy procedure. Mice were anaesthetized with 1–2.5% of isoflurane gas in oxygen and secured on a stereotax (Kopf Instruments) using ear bars. A single large cut was made to cut away the majority of the skin above the skull while attempting to match the 3D-printed frame to the exposed skull. We then used a 3 mm (or 4 mm, depending on the window implant size) biopsy punch to carefully centre the craniotomy over the AAV injection site while avoiding the sagittal suture, and slowly and gently rotated the biopsy punch until the bone could be lifted away with 5/45 forceps. We used saline to irrigate regularly while performing the craniotomy, taking care not to puncture the dura. After stopping any bleeding, we placed a small amount of silicone oil (Sigma-Aldrich 181838) to cover the brain, then carefully placed the stacked window over the brain. We then applied pressure to the top of the stacked window with a thinned wooden applicator (for example, the back of a cotton swab or toothpick) mounted on the stereotaxic arm until the 4 mm (or 5 mm) coverslip was flush with the skull surface. We used cyanoacrylate to glue around the window and waited until it dried before removing pressure from the thinned wooden applicator. We then positioned the headpost over the skull and used C&B Metabond to cement the headpost in place, taking care not to cover the cranial window. After the Metabond dried, we applied some silicone elastomer (World Precision Instruments, Kwik-Sil) over the cranial window to protect it from any damage. We then administered post-operative drugs (meloxicam at 5 mg kg$^{-1}$ and 0.25% bupivicaine around the headpost implant) and allowed the mouse to recover for at least 3 d before imaging.

**Recording sessions.** Before each animal's first awake head-fixed imaging session, the animals were acclimated to head-fixation and the treadmill setup for at least three sessions. Additionally, to minimize stress before awake imaging session, we acclimated the animals to mechanical fixation in which the head was restrained. Animals received chocolate milk (Nesquik) as a reward while running on a custom-built treadmill. Once acclimated, animals were imaged for up to two sessions per day, where each session lasted no longer than 2 h.

For experiments, mice were head-fixed atop a freely moving treadmill during video capture. Epifluorescence microscope images were captured with a sCMOS camera (Kiralux, 5 MP) through a ×4 objective. Treadmill rotation data (synchronized with the camera) was captured during the experiment. During the recording session, a brushing tactile stimulus was applied to the spine region of the mouse with a minimum of 30 s between stimulus events. Bio-FlatScope recording sessions were captured within 30 min of epifluorescence recordings, with exposure time, recording lengths and stimuli timing matching those with the epifluorescence microscope. ROIs in the brain for comparing treadmill activity across epifluorescence and Bio-FlatScope were selected on the basis of high-activity areas extracted from the epifluorescence video.

**Calcium activity cluster identification and analysis.** This analysis was based on 2 min imaging sessions recorded at 20 Hz over the full FOV of the cranial window using both Bio-FlatScope and an epifluorescence microscope with a ×4 objective. The ROIs for cluster analysis were selected by observing a 500 µm × 500 µm region of spontaneous high activity in the Bio-FlatScope reconstructions (Fig. 4a). This same ROI was used for both Bio-FlatScope and epifluorescence data. Figure 4b,c show the clusters extracted through *K*-means as well as the corresponding ΔF/F for these clusters.

The first step to identify the clusters of calcium activity was to separate the smooth, almost constant, global intensity and the sharp variations due to calcium activity. This step was achieved using an optimization method called RPCA[40] that decomposes the time-varying measurements into a low-rank component and a sparse component. The low-rank component embodies the smooth global variations, while the sparse component is from the calcium activity variations. The algorithm used for RPCA is from ref. [41]. The separated calcium activity was then grouped into clusters of similar activity using *K*-means clustering algorithm. This process was done on the reconstructed data from Bio-FlatScope and is shown in Fig. 4.

**Human subjects for oral mucosa imaging.** The in vivo human study was performed at Rice University. Volunteers recruited in the study were 18 years or older, with no underlying health conditions. All volunteers provided health pre-screen information to reduce the risk of Coronavirus disease spread. The study was in accord with the protocol approved by the Institutional Review Board at Rice University. Written informed consent was obtained from all volunteers before imaging.

**In vivo oral mucosa imaging.** Bio-FlatScope imaging and microscopic imaging were performed on each human volunteer. A Bio-FlatScope prototype based on a contour phase mask with 6 µm feature size was used in this experiment. During a Bio-FlatScope imaging session, volunteers were instructed to place their chin and forehead on a custom chin-head rest for maximum stability (Supplementary Fig. 16). Illumination was achieved by a tabletop green LED (X-Cite XLED1) and a liquid light guide (3 mm diameter) mounted in the chin rest with illumination intensity less than 20 mW. The illumination source was used 3–5 min per imaging session, resulting in a maximum illumination intensity at the skin of less than the threshold limit value specified by National Conference of Governmental Industrial Hygienists (ACGIH). During the experiment, the volunteer positioned their lower lip over the tip of the liquid light guide with one hand and placed the imaging device onto the illuminated area with the other hand. The imaging device consists of a Bio-FlatScope prototype and an opaque handheld enclosure, designed to maintain a constant 3 mm working distance, with a 2.5-mm-wide square aperture.

The microscopic imaging session was performed with a microscope objective (Nikon Fluor ×4, NA 0.13 PhL DL). During the imaging session, the volunteer was instructed to position their lower lip with one hand and hold the liquid light guide with the other hand to keep the tip in contact with the bottom side of the lower lip.

**Vessel counting in the human oral mucosa.** Vessel counting was based on images taken by the ×4 objective and reconstructed images from the Bio-FlatScope prototype. A grid containing 2 mm of horizontal lines and 2 mm of vertical lines was superimposed on the image. The capillary density was calculated similar to the method in refs. [42,43]. The number of small vessels (<20 µm) per millimetre was calculated as the number of small vessels crossing the grid, divided by the total length of the grid. Two researchers independently performed vessel counting for each image and the results were averaged. In each volunteer, the images from ten different areas were averaged to give a final capillary density value.

**Statistics and reproducibility.** For Fig. 1d, the imaging of the USAF test target and *Convallaria* with each respective prototype was repeated more than 10 times, with similar results. For Fig. 2a–f, the imaging of fluorescent beads in non-scattering and scattering media was performed 3 times, with similar results. For Fig. 2g, the imaging of the fixed *Hydra* was performed 3 times, with similar results. The image with the most distinguished depth difference across the scene was selected. For Figs. 3 and 4, and Supplementary Figs. 8–10,13,15 and 18: the total number of mice imaged in the experiments is 7 and reconstructions can be made with them. We selected 2 in the Results on the basis of the expression level, surgery performance and mouse activity (shown in Figs. 3 (and Supplementary Figs. 8–10 and 15) and 4 (and Supplementary Figs. 13 and 18)). For Fig. 5 and Supplementary Fig. 14, 5 volunteers were recruited. For each volunteer, 10 images were taken with Bio-FlatScope and ×4 objective, respectively, with similar quality. The images shown were selected on the basis of vasculature pattern and density in the imaged region. For Supplementary Fig. 4, the total number of *Hydra* imaged was more than 5. Multiple video clips were taken with each animal. We selected one video clip on the basis of the expression level and animal activity. For Supplementary Fig. 5, more than 5 mouse brain slice samples were imaged, with similar results. For Supplementary Fig. 6, the experiment was repeated more than 10 times, with similar results. For Supplementary Fig. 11, the simulation result was the same with each run, and the experiment result is the same image as shown in Fig. 1d.

**Reporting Summary.** Further information on research design is available in the Nature Research Reporting Summary linked to this article.

## Data availability

The main data supporting the results in this study are available within the paper and its Supplementary Information. Sample data are available on GitHub at https://github.com/JiminWu/Bio-FlatScope. The raw and analysed datasets generated during the study are too large to be publicly shared, yet they are available for research purposes from the corresponding authors on reasonable request. The raw healthy volunteer data are available from the corresponding authors, subject to approval from Rice University's Institutional Review Board.

## Code availability

Custom MATLAB codes for Bio-FlatScope image reconstruction are available on GitHub at https://github.com/JiminWu/Bio-FlatScope. Other custom MATLAB codes, including Bio-FlatScope capture, simulation and RPCA analysis, are available from the corresponding authors on request.

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

## Acknowledgements
This work was supported in part by DARPA grant N66001-17-C-4012, NIH grant RF1NS110501 and NSF grant IIS-1730574. This research was sponsored by the Defense Advanced Research Projects Agency (DARPA) through Cooperative Agreement D20AC00002 awarded by the US Department of the Interior (DOI), Interior Business Center. The content of the Article does not necessarily reflect the position or policy of the US Government, and no official endorsement should be inferred. We thank K. Badhiwala for the useful discussions on *Hydra vulgaris* neurobiology and behaviour; C. Juliano (UC Davis) and R. Steele (UC Irvine) for sharing transgenic *Hydra*; F. Ye for the scanned electron micrograph of the phase mask and the fibre/microprism part; and J. Juneau for the mounted mouse brain slices.

## Author contributions
J.K.A., D.Y., J.W and V.B. fabricated and characterized prototypes, and designed and built the experimental hardware setup. V.B. developed the reconstruction model, computational algorithms and simulation platform. S.G. prepared experimental mice. J.K.A., A.V.R., S.G. and D.Y. performed the mouse imaging experiment. J.K.A. performed the freely moving *Hydra* imaging experiment. J.K.A. and V.B. analysed the mouse and freely moving *Hydra* imaging data. D.Y. and J.W. performed the human oral mucosa imaging experiment and analysed data. J.C. aided in the human oral mucosa imaging experiment. S.K. prepared fixed *Hydra* samples. J.W. and S.K. performed the fixed *Hydra* imaging experiment. R.R.-K., C.K., A.V. and J.T.R. supervised the research. All authors contributed to the writing of the manuscript.

## Competing interests
J.K.A., A.V., V.B. and J.T.R. are inventors on intellectual property related to this type of lensless imaging.

## Additional information

**Correspondence and requests for materials** should be addressed to Ashok Veeraraghavan or Jacob T. Robinson.

# Reporting Summary

## Statistics

For all statistical analyses, confirm that the following items are present in the figure legend, table legend, main text, or Methods section.

| n/a | Confirmed | |
|---|---|---|
| ☐ | ☒ | The exact sample size (*n*) for each experimental group/condition, given as a discrete number and unit of measurement |
| ☐ | ☒ | A statement on whether measurements were taken from distinct samples or whether the same sample was measured repeatedly |
| ☒ | ☐ | The statistical test(s) used AND whether they are one- or two-sided<br>*Only common tests should be described solely by name; describe more complex techniques in the Methods section.* |
| ☒ | ☐ | A description of all covariates tested |
| ☒ | ☐ | A description of any assumptions or corrections, such as tests of normality and adjustment for multiple comparisons |
| ☐ | ☒ | A full description of the statistical parameters including central tendency (e.g. means) or other basic estimates (e.g. regression coefficient) AND variation (e.g. standard deviation) or associated estimates of uncertainty (e.g. confidence intervals) |
| ☒ | ☐ | For null hypothesis testing, the test statistic (e.g. *F*, *t*, *r*) with confidence intervals, effect sizes, degrees of freedom and *P* value noted<br>*Give P values as exact values whenever suitable.* |
| ☒ | ☐ | For Bayesian analysis, information on the choice of priors and Markov chain Monte Carlo settings |
| ☒ | ☐ | For hierarchical and complex designs, identification of the appropriate level for tests and full reporting of outcomes |
| ☒ | ☐ | Estimates of effect sizes (e.g. Cohen's *d*, Pearson's *r*), indicating how they were calculated |

*Our web collection on statistics for biologists contains articles on many of the points above.*

## Software and code

Policy information about availability of computer code

| | |
|---|---|
| Data collection | MATLAB. Imaging Source IC Capture. Thorlabs Thorcam. |
| Data analysis | The custom MATLAB codes for Bio-FlatScope image reconstruction are available on GitHub at https://github.com/JiminWu/Bio-FlatScope. Other custom MATLAB codes, including Bio-FlatScope capture, simulation and RPCA analysis, are available from the corresponding authors on request. |

For manuscripts utilizing custom algorithms or software that are central to the research but not yet described in published literature, software must be made available to editors and reviewers. We strongly encourage code deposition in a community repository (e.g. GitHub). See the Nature Portfolio guidelines for submitting code & software for further information.

## Data

Policy information about availability of data

All manuscripts must include a data availability statement. This statement should provide the following information, where applicable:

- Accession codes, unique identifiers, or web links for publicly available datasets
- A description of any restrictions on data availability
- For clinical datasets or third party data, please ensure that the statement adheres to our policy

The main data supporting the results in this study are available within the paper and its Supplementary Information. Sample data are available on GitHub at https://github.com/JiminWu/Bio-FlatScope. The raw and analysed datasets generated during the study are too large to be publicly shared, yet they are available for research purposes from the corresponding authors on reasonable request. The raw healthy volunteer data are available from the corresponding authors, subject to approval from Rice University's Institutional Review Board.

# Field-specific reporting

Please select the one below that is the best fit for your research. If you are not sure, read the appropriate sections before making your selection.

☒ Life sciences ☐ Behavioural & social sciences ☐ Ecological, evolutionary & environmental sciences

For a reference copy of the document with all sections, see nature.com/documents/nr-reporting-summary-flat.pdf

# Life sciences study design

All studies must disclose on these points even when the disclosure is negative.

| | |
|---|---|
| Sample size | Human oral-mucosa imaging: Past studies have shown that microvessel-density variation is not significant among healthy human subjects. The use of 5 participants and 10 regions per participant were determined to sufficiently account for the variation of vessel counts within the field of view of each capture, and to show the vessel-counting correlation between 4x objective and Bio-FlatScope. |
| Data exclusions | No data were excluded from the analyses. |
| Replication | We imaged multiple regions of oral mucosa for each participant. Results are consistent for all images taken. |
| Randomization | This is not relevant to the study, as the conclusion of the microvessel-density correlation was drawn by the comparison of data collected from the same participant by two different methods. |
| Blinding | Blinding is not relevant to the study, as the conclusions drawn from the experiments are not dependent on the subject's awareness of being selected. |

# Reporting for specific materials, systems and methods

We require information from authors about some types of materials, experimental systems and methods used in many studies. Here, indicate whether each material, system or method listed is relevant to your study. If you are not sure if a list item applies to your research, read the appropriate section before selecting a response.

### Materials & experimental systems

| n/a | Involved in the study |
|---|---|
| ☒ | ☐ Antibodies |
| ☒ | ☐ Eukaryotic cell lines |
| ☒ | ☐ Palaeontology and archaeology |
| ☐ | ☒ Animals and other organisms |
| ☐ | ☒ Human research participants |
| ☒ | ☐ Clinical data |
| ☒ | ☐ Dual use research of concern |

### Methods

| n/a | Involved in the study |
|---|---|
| ☒ | ☐ ChIP-seq |
| ☒ | ☐ Flow cytometry |
| ☒ | ☐ MRI-based neuroimaging |

# Animals and other organisms

Policy information about studies involving animals; ARRIVE guidelines recommended for reporting animal research

| | |
|---|---|
| Laboratory animals | Hydra vulgaris, transgenetic, expressing either GFP in interstitial cell lineage (GFP, Neurons) or GCaMP7b in ectodermal muscle cells (GCaMP7b, Ecto). <br><br> Wild-type C57BL/6 mice (n = 7) from Charles River Laboratories were used for this study. All animals (n = 7) were injected with the adeno-associated viral vector AAV9.CamKII.GCaMP6f.WPRE.SV40 (Penn Vector Core). |
| Wild animals | The study did not involve wild animals. |
| Field-collected samples | The study did not involve samples collected from the field. |
| Ethics oversight | All experimental procedures were approved by the Institutional Animal Care and Use Committee at Rice University, and followed the guidelines of the National Institute of Health. |

Note that full information on the approval of the study protocol must also be provided in the manuscript.

## Human research participants

Policy information about studies involving human research participants

| | |
|---|---|
| Population characteristics | All human research participants recruited in the study were 18 years or older, with no underlying health conditions (more specifically, not infected by diseases that affect microvessel density in the oral mucosa). |
| Recruitment | Participants were recruited within Rice University. No potential bias may be present that are likely to impact the results of the study, as past studies have shown that microvessel-density variation is not significant among healthy human subjects. |
| Ethics oversight | All volunteers provided health pre-screen information to reduce the risk of COVID spread. The study was in accord with the protocol approved by the Institutional Review Board at Rice University. Written informed consent was obtained from all volunteers prior to imaging. |

Note that full information on the approval of the study protocol must also be provided in the manuscript.

