## [Peer Review File · Nature Biomedical Engineering]

In vivo lensless microscopy via a phase mask generating diffraction patterns with high-contrast contours

Corresponding author: Jacob Robinson

Editorial note

This document includes relevant written communications between the manuscript's corresponding author and the editor and reviewers of the manuscript during peer review. It includes decision letters relaying any editorial points and peer-review reports, and the authors' replies to these (under 'Rebuttal' headings). The editorial decisions are signed by the manuscript's handling editor, yet the editorial team and ultimately the journal's Chief Editor share responsibility for all decisions.

Any relevant documents attached to the decision letters are referred to as **Appendix #**, and can be found appended to this document. Any information deemed confidential has been redacted or removed. Earlier versions of the manuscript are not published, yet the originally submitted version may be available as a preprint. Because of editorial edits and changes during peer review, the published title of the paper and the title mentioned in below correspondence may differ.

Correspondence

Fri 05 Mar 2021

Decision on Article nBME-21-0046

Dear Dr Robinson,

Thank you again for submitting to *Nature Biomedical Engineering* your manuscript, "*In vivo* imaging with a flat, lensless microscope", and please excuse the delay. The manuscript has been seen by 3 experts, whose reports you will find at the end of this message, however, despite our chasing efforts, one reviewer was unable to provide comments. You will see that although the reviewers have some good words for the work, they articulate concerns about the degree of advance that this study represents over your previous published work, and in this regard provide useful suggestions for improvement. We hope that with further work you can address the criticisms, increase the level of significance of the study, and convince the reviewers of its merits. In particular, we would expect that a revised version of the manuscript provides:

* an improved performance analysis of the Flatscope 2.0 in vivo, including the robustness and reproducibility of the data, as well as an improved description of the phase mask.

* a restructured manuscript that focuses on the in vivo use of the Flatscope 2.0, in mammalian systems (including the human data) and clearly illustrates its advantageous use, over the state-of-the-art.

When you are ready to resubmit your manuscript, please upload the revised files, a point-by-point rebuttal to the comments from all reviewers, the (revised, if needed) reporting summary, and a cover letter that explains the main improvements included in the revision and responds to any points highlighted in this decision.

Please follow the following recommendations:

* Clearly highlight any amendments to the text and figures to help the reviewers and editors find and understand the changes (yet keep in mind that excessive marking can hinder readability).- * If you and your co-authors disagree with a criticism, provide the arguments to the reviewer (optionally, indicate the relevant points in the cover letter).
- * If a criticism or suggestion is not addressed, please indicate so in the rebuttal to the reviewer comments and explain the reason(s).
- * Consider including responses to any criticisms raised by more than one reviewer at the beginning of the rebuttal, in a section addressed to all reviewers.
- * The rebuttal should include the reviewer comments in point-by-point format (please note that we provide all reviewers will the reports as they appear at the end of this message).
- * Provide the rebuttal to the reviewer comments and the cover letter as separate files.

We hope that you will be able to resubmit the manuscript within 25 weeks from the receipt of this message. If this is the case, you will be protected against potential scooping. Otherwise, we will be happy to consider a revised manuscript as long as the significance of the work is not compromised by work published elsewhere or accepted for publication at *Nature Biomedical Engineering*. Because of the COVID-19 pandemic, should you be unable to carry out experimental work in the near future we advise that you reply to this message with a revision plan in the form of a preliminary point-by-point rebuttal to the comments from all reviewers that also includes a response to any points highlighted in this decision. We should then be able to provide you with additional feedback.

We hope that you will find the referee reports helpful when revising the work. Please do not hesitate to contact me should you have any questions.

Best wishes,

Rosy

Dr Rosy Favicchio
Senior Editor, Nature Biomedical Engineering

Reviewer #1 (Report for the authors (Required)):

J Adams et al Bio-Flatscope paper review

This paper demonstrates the use of miniaturized lensless imaging systems for in vivo imaging. The advance, over the previous work from these groups (Adams et al Sci Adv) has been the implementation of a phase mask, that allows imaging in low-light and low contrast settings. Taking advantage of these new capability, the authors present three sets of data - whole animal 3D fluorescent imaging of *Hydra vulgaris*, imaging surface of head-fixed, awake mice expressing GCaMP, and demonstrate clinical application imaging human oral mucosa imaging.

This is an interesting development in the use of lensless, semi-miniaturized device for in vivo imaging. There are improvements in the imaging capabilities of lensless systems compared to previous version. The data provided currently, is however not sufficient to convince me that this is a robust technology that is ready to be widely implemented for in vivo biological imaging. Second, I am not yet clear what new experimental or imaging capabilities this technology provides in comparison with existing imaging modalities.

In principle, the bio-flatscope should enable (i) large field of view (FOV), (ii) near cellular resolution, (iii) fast, (iv) 3D imaging in a (v) integrated miniaturized device. The system does achieve large FOV (30x compared to mini scopes, although the comparison should really be with a conventional epi-fluorescence microscope), but the resolution of the device is poor. The in vivo data presented in mice is mesoscopic imaging. Convincing 3D reconstruction is presented in only hydra at a rather slow speed of 2Hz speed, suggesting

there are still limitations to the technology (masking requires high exposure times to discern dim signals?). Based on the data presented, it will be difficult to apply this technology for 3D imaging in scattering tissue. The system architecture intrinsically allows miniaturization, potentially enabling imaging studies in freely behaving animals performing complex behaviors. However, in its current implementation the device is fairly large and works only in head restrained mice. While I understand a new technology need not check all these boxes, as presented in this manuscript, the in vivo imaging capabilities fall short on several aspects, and this, in my opinion, limits practical applicability and adoption by the broader community in its current iteration.

With regards to my second comment, with the exception of 3D imaging of the hydra, all other imaging demonstrations – in mice as well as in humans can be performed with existing technologies that are as miniaturized as the bio-flatscope with better imaging performance.

In summary, this is a technical improvement in the sense that the lensless flatscope format device can now perform imaging in low light, low contrast conditions, but needs further optimization before it can be practically implemented for in vivo 3D biological imaging.

Below, I describe my major and minor concerns:

Major Concerns/ questions

Hardware:

1) The use of the micro prisms coupled to optic fibers is interesting. However, there is no quantification on how well such illumination works. How does illumination vary across the FOV? This is important in the context of dynamic imaging. For instance, if the hydra moves between areas that are dimly lit, as compared to well lit areas, the fluorescence signal is going to vary.

Insufficiency of presented data:

2) Mice experiments: The reporting summary states $n = 3$ animals, which seems like the bare minimum. There are several factors to consider in the design of these experiments - efficacy of viral injections, variation in expression levels of GCaMP, variation in GCaMP activity from mouse to mouse etc, which necessitates higher number of animals. Further, Figure 3 and 4 show representative data from a single mouse and statistical analysis is performed on the data from the single mouse.

How does the performance of the bio-flatscope vary when taking into consideration all the biological variables?

- Can authors provide reconstructions from multiple animals?
- What is the average signal to noise ratio obtained in multiple mice?

3) Hydra imaging: Calcium imaging data is provided from just one hydra vulgaris experiment.

For the non-hydra experts, it is difficult to make sense of the data provided. Are the Ca signals shown typical for certain types of 3D body shapes of the organism? Are they consistent across multiple hydra imaged?

What unique insight about hydra behavior or activity as imaged by calcium imaging is being revealed by the bio-flatscope? With the representative data provided it is currently not possible answer these questions.

Are the slow drifts down in DFF after the “bursts” simply an artifact of the organism drifting in and out of the plane? Or perhaps changing its relative orientation with respect to the bio-flatscope results in changes to the DFF measured?

How does this compare with imaging performed using conventional instruments? Without a proper benchmark, e.g., with a conventional light-field imaging system, it is difficult to ascertain how well 3D imaging really work. While 3D imaging of freely floating hydra may be difficult to perform using conventional imaging setup, can an experiment be performed with restrained hydra that allows a comparative study?

Depth resolution:

4) What is the depth resolution of the bioflatscope in (i) non-scattering medium (ii) scattering media? Some data needs to be provided perhaps using static 3D phantoms with and without scattering properties, as is commonly done. E.g., See Prevedel* et al Nature Methods Figure 1d-g. This is very important to characterize prior to performing dynamic imaging in vivo.

5) 3D reconstruction is shown in Fig.2 in hydra. The data shown in Supplementary Fig. 11 when imaging the mouse brain surface does not appear to show any discernible difference between the different depths. I am currently not convinced with the data provided that 3D dynamic imaging is possible, particularly in scattering tissue.

6) “Here the large FOV and 3D reconstructions help us capture movies of whole animal calcium dynamics, which can be used by neuroscientists to study information processing in these small model organisms” Expanding on this point may help the reader ascertain what kind of information processing one may perform. At present, the resolution seems to be too limited to image single cells, or even mesoscale flow of calcium activity within the organism.

Benchmarking against existing epifluorescence/lightfield imaging system:

7) At present, this paper lacks a clear and quantified benchmarking of the flatscopes of depth resolution. This could be addressed by perhaps showing (i) comparative 3d imaging in hydra using a conventional or light field imaging scope, (ii) comparative widefield mesoscopic imaging in mice using a conventional epifluorescence imaging setup.

Lack of in vivo data showing depth scanning:

In my opinion, the primary advantage a flatscope style device is it's the ability to perform 3D imaging. Several questions arise that need clarification.

8) From the data provided, it looks like the Hydra Vulgaris imaging was done in 3D, while imaging in mice and humans was performed in 2D. While supplementary data shows image reconstructions at different depths from the sensor, the different reconstructions appear mostly similar -whereas, typically one should be able to trace 3D blood vessel geometry with even low magnification widefield objectives. Are there any differences in the mesoscale activity maps from different depths?

9) Conventional light field imaging suffers from poor resolution in z direction. What is the depth resolution obtainable with the bio-flatscope?

10) How feasible is 3D imaging in scattering tissue?

11) How does resolution and image intensity vary as a function of depth of reconstruction? Figure 1 in the manuscript provides data from planar samples. Some data needs to be provided perhaps using static 3D phantoms with and without scattering properties, as is commonly done. E.g., See Prevedel* et al Nature Methods Figure 1d-g. This is very important to characterize prior to performing dynamic imaging in vivo.

12) Does image reconstruction result in artifacts? I.e., if a sample with a fixed fluorescence pattern is imaged and reconstructed when located at different distances from the bio-flatscope, how does the intensity vary? The authors provide resolution data. But changes in intensities if any need to be quantified. If the intensity varies, then quantifying DF/F in a moving object such a hydra would be prone to artifacts arising due to image reconstruction.

In vivo mouse imaging:

13) Calcium traces are currently shown over very long lengths scale. For clarity, zoomed in view of a few seconds of data should be provided for both flatscope and epi-fluorescence data. Without that data, it is difficult to qualitatively and quantitatively compare the data shown in Figure. 3.

14) The clusters shown in Fig. 5 are very large - >100 micrometers in some cases. This will have several cells within, with a few cell's dominating the responses measured. So the claim of 'near cellular resolution' is a stretch.

Methods:

15) In contrast with traditional single photon imaging, data here is reconstructed. There is little detail on how the reconstructed images were analyzed. It is possible the analysis methodology proceeds along similar lines to say data obtained from mini scopes, but for the sake of completeness, this information should be provided. For instance, I had several questions for which I did not get an answer in the methods section.

16) Since images can be reconstructed at multiple planes, what plane was chosen for reconstruction? The surface of the brain or a plane slightly underneath?

17) If image reconstruction was from underneath the surface, how far down could images be reconstructed? What are the limitations to such 3D reconstructions? I think this is important to discuss, and will allow the reader to ascertain the limitations of the technology for in vivo imaging.

- Currently, the reconstructions are shown as distance from the sensor, which is not informative.

-How is data analyzed once a reconstructed image is obtained?

-What does DF/F signify in Fig. 2C? Is this the change in DF/F over the whole FOV? Is this the change in DF/F over the whole FOV averaged across multiple z-planes after reconstruction. Or is it simply DF/F for the whole raw image?

18) How does reconstruction algorithm performance change as a function of depth of reconstruction? The supplementary data provided discusses resolution. But for dynamic In vivo imaging, it is important to quantify changes in intensity as well, especially if the end application is to perform imaging in whole moving organisms.

Minor concerns

1) There is a lack of detail on the hardware of the bio flatscope. More information will be helpful.

(a) What is the function of the interference filter?

(b) The use of the absorptive filter is interesting. How well does it work? Is there a particular need to use an absorptive filter as against a regular glass filter?

For the sake of completeness, this information should be provided.

2) Since the data shown in mice is changes in fluorescence in bulk tissue. It will be prone to artifacts arising from hemodynamic changes. This should be acknowledged.

3) In the supplementary table, the authors compare the performance of the bio-flatscope to other miniaturized imaging devices. In its current form, the bio-flatscope is still too bulky and all the experiments performed by the authors are on head fixed animals. The comparison need to be made with respect to a conventional wide field epi-fluorescence scope or a light-field imaging setup.

4) Scale bar would be helpful in Figures 1a and 3a.

Reviewer #2 (Report for the authors (Required)):

The article "In vivo imaging with a flat, lensless microscope" by Adams et al. presents an interesting and exciting extension of the group's previous work on the FlatScope (Ref 10. in the manuscript) to live, fluorescent labeled and unlabeled specimens for the first time in a low-profile, compact lens-less microscope. They utilized a specialized phase mask optimized for dense, low contrast fluorescent samples. Several exciting applications were demonstrated such as on fluorescent test samples (USAF and mouse brain tissue), live samples such as the *Hydra vulgaris* (with some limited 3D resolution), and perhaps most exciting with live mounted imaging of mice brains during locomotion and of human oral mucosa imaging. While the work is valid and exciting, in the current instantiation it appears more incremental and not of as high impact as other articles in Nature Biomedical Engineering. Therefore, at the current time I cannot recommend publication in the journal without addressing the following concerns and issues.

In the opinion of this reviewer, the most novel and interesting aspects of the current manuscript are the

application of the FlatScope (Ref. 10 in the manuscript) to live fluorescent labeled tissues and human mucosa. This is apparently a first for lens-less imaging technologies that have multiple advantages such as the small form factor, light weight, and less invasive live biological imaging as demonstrated with the in vivo calcium imaging in mice brains and human mucosal region imaging. Surely the human subjects would feel more comfortable and less worried to have a small chip akin to something like a modern dental x-ray detector put near their mouth than a bulky optical microscope. However, the impact and the novelty of the work is brought into question by numerous other demonstrations of live mouse brain imaging while allowing animal locomotion such as the work by Ziv et al. *Nature Neuroscience* 16, 264 (2013). Furthermore, other groups have demonstrated low profile, lens-less, even single shot fluorescent imaging cameras such as Antipa et al. *Optica* 5, 2334-2536 (2018). Furthermore, the demonstrated single-shot 3D imaging of the Hydra, has comparable but less convincing results as compared to other published results such as Yanny et al, *Light Sci Appl* 9, 171 (2020), although this group did not do in vivo imaging. Furthermore, in my opinion, the demonstration of the 3D imaging of the hydra is not convincing and appears more like a 3D extrusion of a 2D image. However, it is difficult to determine this with the current images. More images and analysis, especially along the third dimension would allow more careful and accurate assessment.

Finally, this leads me to an additional final concern about the manuscript in that the authors attempt to show so many example applications (each of which is exciting and valid), but it reduces the amount of information that can be placed in the main body and dilutes the impact of each of the applications. I would recommend on focusing on the two most transformational and impactful applications for *Nature Biomedical Engineering* which in my opinion are the live mouse brain imaging and human oral mucosa imaging. These applications most clearly highlight the potential clinical applications due to its compactness, light weight, and lens-less nature. Below I list other technical items that should be addressed:

Major technical criticisms or questions:

- 1) The current manuscript is extremely dense and spends too little time on too many applications. I would recommend moving some of the information to the SI (or possibly a separate article) such as the 3D demonstration with the hydra vulgaris. To me, this demonstration does not support the main tenants of the paper and feels like a distraction. This on top of the fact that the claim to 3D resolution seems weak with the current small sub-figures in Figure 2a. It is difficult for me to access the 3D image quality or resolution in the third dimension from the current Fig 2, especially when comparing the top and bottom images. Other information that could be moved to the supplemental information is the description of the mouse brain phantom (it is not even discussed in the main body but in the SI now), and perhaps Fig 3g as I am not sure exactly what it depicts. Furthermore, what resolution is claimed in the 3rd dimension and how is this possible with a relatively flat phase mask compared to for example Antipa et al. *Optica* 5, 2334-2536 (2018) work.
- 2) There is not enough information currently to evaluate the phase mask used. What are the spatial frequencies encoded in the phase mask and the larger range focus characteristics (beyond the few mm range shown in Fig S10 and S11).

Minor technical criticisms or questions:

- 1) How is the working distance or sample to BioFlatScope distance adjusted? Additional figures or description of this, at least in the SI would be helpful. What is the effective depth of focus?
- 2) How sensitive is this working distance to the reconstruction algorithm?
- 3) How computationally intensive is the reconstruction algorithm (single processor/multiple processor, compute time, array size of data, etc)? These details would help the reader evaluate the computational cost of, for example, the movies in the supplemental information.
- 4) Fig. 2 and Page 5, what volume of fluid is the hydra vulgaris polyp confined to? Additional details about this experiment would be helpful as it is obvious from the movie that the sample is tumbling freely, but it is not clear why it leaves the image and then returns. Does this point to some limited 'depth of focus'?
- 5) In Fig 5b, there are a number of vertical and horizontal artifacts in the oral mucosa images as compared to the 4x objective microscope images. We assume these are from computational image reconstruction algorithm, but they should be addressed and discussed. Is there a way to mitigate or remove them? Can they not be addressed with appropriate background subtraction or white field removal?
- 6) In the methods section, line 291-293, a description of the range of depth over which the calibration is performed is given, but no distances are mentioned. Please mention those distances.
- 7) The algorithm described on lines 308-311 for reconstruction of 3D volumes seems to treat each layer independently. This seems like a poor assumption for dense tissue where multiple scattering effects could be important from subsequent layers. Is there a way to address this or is it not important for true 3D imaging in tissues from lens-less imaging modalities?

- 8) Additional text describing how the MTF and resolution is determined/measured in the supplemental Fig S1 and S5 would be helpful. The figure captions are not sufficient.
- 9) Moving Fig. S3 earlier in the draft, perhaps near figure 1, would help the reader to understand the exact geometry of the illumination.

Tue 29 June 2021

Decision on Article nBME-21-0046A

Dear Dr Robinson,

Thank you for your revised manuscript, "*In vivo* imaging with a flat, lensless microscope", which has been seen by the original reviewers and one more expert. In their reports, which you will find at the end of this message, you will see that the reviewers acknowledge the improvements to the work and raise a few additional technical criticisms that we hope you will be able to address. In particular, we would expect that the next version of the manuscript provides:

- * improved characterisation of the Bio-FlatScope's *in vivo* performance, including axial resolution and illumination parameters;
- * improved technical accuracy of the *in vivo* image resolution claimed, please avoid using ambiguous language (such as "approaching cellular resolution"), also expanding the description of the RPCA component of the study; please consider creating a table, for the data shown in Fig 4, describing, for each ROI, its size and its diameter (pixels and μm).
- * please consider contextualising the *in vivo* image performance claims to include parameters that define digital image resolution, such as the Nyquist limit, dynamic range, and SNR; please consider referring to the *in vivo* resolution as the "post-processing" digital image resolution, clarifying that the value describes specifically the mice/samples imaged in this study.
- * the *in vivo* image resolution could be further discussed in a limitations paragraph within the discussion where signal intensity, autofluorescence and filters can be used to list criteria that could be optimized to improve the performance of the Bio-FlatScope.

As before, when you are ready to resubmit your manuscript, please upload the revised files, a point-by-point rebuttal to the comments from all reviewers, the reporting summary and a cover letter that explains the main improvements included in the revision and responds to any points highlighted in this decision.

As a reminder, please follow the following recommendations:

- * Clearly highlight any amendments to the text and figures to help the reviewers and editors find and understand the changes (yet keep in mind that excessive marking can hinder readability).
- * If you and your co-authors disagree with a criticism, provide the arguments to the reviewer (optionally, indicate the relevant points in the cover letter).
- * If a criticism or suggestion is not addressed, please indicate so in the rebuttal to the reviewer comments and explain the reason(s).
- * Consider including responses to any criticisms raised by more than one reviewer at the beginning of the rebuttal, in a section addressed to all reviewers.
- * The rebuttal should include the reviewer comments in point-by-point format (please note that we provide all reviewers will the reports as they appear at the end of this message).
- * Provide the rebuttal to the reviewer comments and the cover letter as separate files.

We hope that you will be able to resubmit the manuscript within 12 weeks from the receipt of this message. If this is the case, you will be protected against potential scooping. Otherwise, we will be happy to consider a revised manuscript as long as the significance of the work is not compromised by work published elsewhere or accepted for publication at *Nature Biomedical Engineering*. Because of the COVID-19 pandemic, should you be unable to carry out experimental work in the near future we advise that you reply to this message with a revision plan in the form of a preliminary point-by-point rebuttal to the comments from all reviewers that also includes a response to any points highlighted in this decision. We should then be able to provide you with additional feedback.

We look forward to receive a further revised version of the work. Please do not hesitate to contact me should you have any questions.

Best wishes,

Rosy

Dr Rosy Favicchio
Senior Editor, Nature Biomedical Engineering

Reviewer #1 (Report for the authors (Required)):

Please see attached pdf document.

Reviewer #2 (Report for the authors (Required)):

The revised article “In vivo imaging with a flat, lensless microscope” by Adams et al. shows substantial improvement over the previous version. I thank the authors for refocusing the emphasis of the article on the live fluorescent labeled tissues and human mucosa. The additional experiment of the suspended fluorescent particles to demonstrate the 3D resolution in clear and scattering medium was also a wonderful addition and helped to prove the power and applicability of the technique, especially to scattering media like live tissue. Improvements to the text include moving some of the information and examples to the supplement, adding the additional fluorescent bead resolution demonstration and simulation, and clarifying many parts of the methods section and supplement. I especially appreciated the additional table information comparing BioFlatScope to other comparable techniques and technologies. One aspect of the BioFlatScope that I believe is most impactful and applicable is the high axial resolution demonstrated in scattering media compared to other techniques combined with a large FOV. I have no additional major concerns about the manuscript and believe it is appropriate for publication in Nature Biomedical Engineering.

However, I would ask that the authors add a few additional details to the text to help clarify some points or add needed information:

- 1) In the response to reviewers on page 20, you state “As Bio-FlatScope reported in this manuscript was designed to work at a relatively narrow working distance range (~1 mm)...” but then later state “The working distance ... for this specific Bio-FlatScope prototype is ~4mm.” This was confusing to me, can you please clarify? Is the working distance ~ 1mm or ~4mm? This does not need to be addressed in the manuscript text, just in the review response.
- 2) The author’s provided a nice description of the computational needs and performance of the algorithm on page 21 of the reviewer response, however it is not included in the methods section of the main manuscript. I would request that it be included in the manuscript or supplement.
- 3) While I appreciate the inclusion of the description about the phase mask in the supplement section 1 (lines 709-719), I was still left for more details about the fabrication of the phase mask. What resist was used, how was etching performed, what depths of etching or etching parameters (temperature, etc)? A basic recipe or additional details about the phase mask manufacturing should be included or referenced.

Reviewer #4 (Report for the authors (Required)):

This paper proposes and prototypes a compact lens-less imaging device for in vivo fluorescence imaging, and conducts actual in vivo fluorescence imaging using the prototype device. The authors clarify the basic characteristics of the proposed device and show that it is useful for in vivo fluorescence imaging. The concept is different from that of a head-mounted miniature fluorescence microscopy system, and this

reviewer judges that this is a significant paper that demonstrates a new application of lens-less imaging.

However, under constrained conditions, the in vivo fluorescence imaging system can be realized with existing microscope systems, and the reviewer finds it difficult to argue the superiority of the proposed method. The reviewer requests the authors to indicate whether the proposed method can be applied to free-living rodents and, if so, please describe the reason.

In addition, there are some points to be considered in this method, such as the following. The reviewer requests the authors to respond to these points.

(1) Excitation light source: The excitation light source layout is shown in Fig. S3, but the excitation light irradiation range seems to be narrow in this layout. Is there any way to widen the excitation range?

(2) In this paper, the authors used a commercially available CMOS image sensor, which has a glass cover, so there is a certain distance between the phase mask and the imaging surface. Does this distance affect the spatial resolution of lens-less imaging?

(3) It is difficult to prevent autofluorescence from the absorption filter by using only two types of filters, absorption filter and interference filter (refer to the following paper

K. Sasagawa et al., "Highly sensitive lens-free fluorescence imaging device enabled by a complementary combination of interference and absorption filters," *Biomedical optics express*, vol.9, no.9, pp.4329-4344, 2018.)

Please comment on this point.

Sat 7 Aug 2021

Decision on Article nBME-21-0046B

Dear Dr Robinson,

Thank you for your revised manuscript, "*In vivo* imaging with a flat, lensless microscope". Having consulted with the original reviewers, I am pleased to write that we shall be happy to publish the manuscript in *Nature Biomedical Engineering*

We are now performing detailed checks on your paper and will send you a checklist detailing our editorial and formatting requirements in due course. Please do not upload the final files until you receive this additional information from us.

Best wishes,

Rosy

Dr Rosy Favicchio
Senior Editor, Nature Biomedical Engineering

Reviewer #1 (Report for the authors (Required)):

I went through the manuscript. The author's new edits address my concerns.

Reviewer #2 (Report for the authors (Required)):

Dear Editors,

I have reviewed again the article "In vivo imaging with a flat, lensless microscope" by Adams et al. The authors have addressed all of my concerns, and I am ready to endorse it for publication in *Nature Biomedical Engineering*. It is an exciting and impactful advance in the field of lensless imaging in being the first application of the small format lensless microscope to living organisms. It will have broad impact in many areas of biomedical imaging and medicine.

Reviewer #4 (Report for the authors (Required)):

My comments have been addressed. I am happy to support the publication of this manuscript in its current version.

Appendix 1

Jesse Adams revised manuscript review

The revised manuscript has addressed most of my comments from earlier. But there are a few remaining concerns. My major concern is still the poor quality data in vivo mouse imaging data presented in the manuscript and the unsubstantiated claims the authors make based on this.

Major concerns

Axial resolution and focusing during in vivo functional imaging: A lot of paper describes 3D refocusing. Based on the data now shown in the revised Figure 2, the axial resolution in scattering medium is 80 micrometers. This is in test samples and is rather poor. The data shown in the in vivo functional imaging will have further confounding factors that will make the axial resolution worse. Indeed the qualitative data shown in Figure S13 shows no discernible change in sharpness of the image captured over 300 micrometers. There is some data shown

- Can the authors please provide an analysis FWHM of the intensity over a line drawn across any of the major blood vessels imaged in the montage S13A? I suspect there will be very little change in FWHM measurements. This will provide a true qualitative indication of the actual focusing capabilities during in vivo functional imaging experiments. This limitation really needs to be acknowledged and discussed in the main manuscript. The current claim of resolution really is in test, fixed samples, and this should be distinguished from the actual resolution obtained in functional imaging experiments.

“This ability is particularly useful for neural imaging as it can be used to account for motion and curvature of the brain”. Correction for motion, particularly when the axial resolution is poor is highly unlikely and non trivial. This is a speculative sentence and not substantiated by any data provided. This part of the sentence needs to be removed.

Functional mouse imaging data: The revised manuscript has still not addressed the issue of benchmarking with respect to a conventional scope as I had commented in my previous review. Given the poor axial resolution in scattering medium, a comparison can simply be made between the flat scope and the episode-fluorescence imaging shown in D.

- For the clusters that are identified, can the authors please provide spontaneous DFF traces for both the data captured using the epifluorescence microscope and the flatscope? This will provide the readers with a better comparison between the two modalities and help benchmark the data.

"The calcium activity in the smallest of these ROIs (10-20 μm diameter) show unique temporal dynamics suggesting that calcium dynamics might be recovered with near cellular resolution." A number of issues with this .

- None of the clusters that the authors show traces for are anywhere close to 10-20 micrometers. The smallest cluster highlighted in the traces are 6 and 3. Both appear to be ~30-40 micrometers in diameter. The trace corresponding to 3 is noisy in both Fig. 4c and Fig. 4f.
- The 'cell like activity claim' hinges on this figure. But there is no reference to data presented in Fig. 4C, Fig. 4F and Fig 4e anywhere in the main manuscript.
- The processed data in Fig.4f looks noisy, particularly for ROIs 2 and 3.

This claim is really a stretch and should be removed.

Minor Comments

The new Figure 2 as inserted in the manuscript pdf is pixelated.

A number paragraphs in the results section of the paper e.g., lines 199-214, lines 236-245 should really be in the discussion section?

Rebuttal 1

Editorial comments:

We hope that with further work you can address the criticisms, increase the level of significance of the study, and convince the reviewers of its merits. In particular, we would expect that a revised version of the manuscript provides:

* an improved performance analysis of the Flatscope 2.0 in vivo, including the robustness and reproducibility of the data, as well as an improved description of the phase mask.

* a restructured manuscript that focuses on the in vivo use of the Flatscope 2.0, in mammalian systems (including the human data) and clearly illustrates its advantageous use, over the state-of-the-art.

We thank the editor for gathering these reviews and the opportunity to improve this manuscript by addressing them. Below we summarize the major changes, provide a list of new experiments and analysis, followed by a point-by-point response to the reviewers:

Summary of Major Changes:

1. **Improved performance analysis:** We added new experiments to characterize the 3D imaging ability of Bio-FlatScope and demonstrate improved single-shot 3D reconstruction of *Hydra* (see **Experiment 1** and revised Figure 2 below). We also characterized how Bio-FlatScope performs in the presence of scattering and non-uniform illumination (**Experiment 1 and Simulation 1**, Figures 2c&d and Figure S7), and demonstrate computational refocusing in images of the human oral mucosa (**Analysis 1**, Figure S14). Finally, we improved our description of the phase mask in section 1 of the supplementary information.
2. **Restructuring of the manuscript to focus on advantages of Bio-FlatScope for mammalian imaging:** As suggested by the reviewers we reorganized the manuscript by moving the epifluorescence calcium imaging of whole animals with *Hydra vulgaris* to the supplementary. We further revised the text to focus on *in vivo* imaging of mammalian systems, and the advantages of Bio-FlatScope. Specifically, we revised Figure 2 of the manuscript to compare the performance of Bio-FlatScope to a confocal microscope, and added **Table S2** to the manuscript, which compares Bio-FlatScope with state-of-the-art lensless imaging technologies. This comparison also shows that Bio-FlatScope features the largest FOV and is the ***first lensless imaging device that has demonstrated the ability to image mammalian tissue.***

List of additional experiments, simulations and analysis:

Experiments:

Experiment 1 (Reviewer 1,2)

(included in the revised manuscript as line 106-131 and Fig. 2)

To characterize the 3D imaging ability, we prepared a 3D test sample by suspending 10 μm fluorescent beads in a $\sim 3 \times 3 \times 0.5 \text{ mm}^3$ clear phantom. For ground truth, we captured the 3D fluorescent phantom using a scanning confocal microscope (Revised Fig. 2a). We find excellent agreement between the Bio-FlatScope reconstruction (Revised Fig. 2a) and the confocal data over a depth range of 500 μm . Empirically, we found that the full width at half maximum (FWHM) of the axial spread of the 10 μm beads is approximately 50 μm in the Bio-FlatScope reconstructions (Revised Fig. 2b), which indicates a 50 μm axial resolution. Next, to evaluate the performance of Bio-FlatScope in scattering media, we prepared a 3D sample using the same protocol as the clear phantom, but with the addition of 1 μm nonfluorescent polymer beads to mimic the scattering property of tissue [1,2]. The scattering strength is controlled by the density of the polymer beads, and we chose the scattering coefficient similar to mouse brain tissue. The FWHM of the axial spread of the 10 μm beads is approximately 80 μm in the Bio-FlatScope reconstructions (Revised Fig. 2e), which indicates an 80 μm axial resolution in scattering media.

To better demonstrate that Bio-FlatScope is capable of single-shot 3D reconstruction of biological samples we imaged a fixed *Hydra vulgaris* expressing GFP in the ectoderm with Bio-FlatScope and compared the reconstructions to a confocal stack as ground-truth. Here, *Hydra* samples were captured by Bio-FlatScope immediately after being prepared. Ground truth images of *Hydra* samples were then taken using a confocal microscope immediately after Bio-FlatScope imaging. The reconstruction of Bio-FlatScope clearly indicates the depth information of the *Hydra* closely corresponds to the ground truth image (Revised Fig. 2g).

Revised Fig. 2. 3D volume reconstruction of fixed fluorescent samples. (A) Bio-FlatScope reconstruction in non-scattering media as a maximum intensity projection along z-axis as well as a ZY slice and an XZ slice compared with the ground truth captured by confocal microscopy (10× objective). Scale bar 100 μm. (B) Axial (XZ) profile and corresponding X and Z profiles of a reconstructed bead in non-scattering media, showing lateral and axial resolution, respectively. Scale bar 20 μm. (C) Bio-FlatScope reconstruction in scattering media as a maximum intensity projection along z-axis as well as a ZY slice and an XZ slice compared with the ground truth captured by confocal microscopy (10× objective). (D) Axial (XZ) profile and corresponding X and Z profiles of a reconstructed bead in scattering media, showing lateral and axial resolution, respectively. (E) Estimated 3D positions of the beads from the Bio-FlatScope reconstruction in non-scattering media, compared with the ground truth captured by confocal microscopy (10× objective). Scale bar 100 μm. Zoom in shows a 3D volume of a selected area, scale bar 20 μm. (F) Estimated 3D positions of the beads from the Bio-FlatScope reconstruction in scattering media, compared with the ground truth captured by confocal microscopy (10× objective). (G) Bio-FlatScope reconstruction of a fixed *Hydra* sample, compared with the ground truth captured by confocal microscopy (10× objective). Scale bar 500 μm.

Simulations:

Simulation 1 (Reviewer 1)

(included in the revised manuscript as Fig. S7)

To characterize how variations in illumination intensity may affect BioFlatScope reconstructions, we simulated the reconstruction of a target with non-uniform irradiance across the entire area (to mimic a fluorescence target under non-uniform illumination). The reconstruction results clearly indicate a linear relationship between illumination intensity and reconstruction intensity, which will allow renormalization of fluorescence intensity as is traditionally done for fluorescence microscopy [3].

Fig S7. Simulation results of fluorescent beads with non-uniform irradiance. a. Simulated target. Ten fluorescent beads in total (3×3 pixels, each pixel corresponding to 4 μm length as consistent with our experimental determined pixel size, i.e. each bead is 12 μm large) with different illumination intensities (hence different irradiance) are simulated as the imaging target. b. Reconstruction result of the simulated target. c. Intensity plot of the reconstructed beads under different illumination intensities. We can see that the intensity of each reconstructed bead is linearly proportional to the intensity of illumination applied to it.

Simulation 2 (Reviewer 1)

To ensure that small variations in the sample depth do not produce changes in the reconstructed fluorescence intensity we simulated the reconstruction of a fluorescent bead with the same irradiance, but at different depths, ranging from 3 mm to 4.8 mm. The intensity when imaging depth extended to 4.8 mm, the reconstruction intensity still remains 72.7% of the maximum intensity. The decrease of reconstruction intensity was caused by lower light collection efficiency at lower depth.

Fig. Reconstruction intensity of a fluorescent bead with the same irradiance at different depths.

Analysis:

Analysis 1 (Reviewer 1,2)

(included in the revised manuscript as Fig. S14)

We reconstructed our human oral mucosa images with PSFs at different depths to demonstrate the computational refocusing ability of Bio-FlatScope. The two images shown below are reconstructed from the same raw capture, with two different PSFs with a depth separation of 180 μm .

Fig S14. Computational refocusing in human oral mucosa. (A) Bio-FlatScope reconstructions at 2.88mm depth. The image is out of focus and some of the small features cannot be reconstructed. (B) Bio-FlatScope reconstructions at 3.04mm depth. The image becomes sharp and small vessels can be reconstructed at high contrast. Scale bar, 100 μ m

Materials added and edited:

Figures and tables:

1. Added Fig. 2 (new), S7, S14, S17.
2. Added Table S2.
3. Moved Fig. 2 (old) from main text to supplemental information as Fig. S4 and corrected the time scalebar from 30 s to 15 s (was a typo).

Text:

1. Text edits are marked in the revised manuscript as a different color.
2. The *Hydra* paragraphs are moved from main text to supplemental information.

Reviewer #1:

J Adams et al Bio-Flatscope paper review

This paper demonstrates the use of miniaturized lensless imaging systems for in vivo imaging. The advance, over the previous work from these groups (Adams et al Sci Adv) has been the implementation of a phase mask, that allows imaging in low-light and low contrast settings. Taking advantage of these new capability, the authors present three sets of data - whole animal 3D fluorescent imaging of *Hydra vulgaris*, imaging surface of head-fixed, awake mice expressing GCaMP, and demonstrate clinical application imaging human oral mucosa imaging.

This is an interesting development in the use of lensless, semi-miniaturized device for in vivo imaging. There are improvements in the imaging capabilities of lensless systems compared to previous version. The data provided currently, is however not sufficient to convince me that this is a robust technology that is ready to be widely implemented for in vivo biological imaging. Second, I am not yet clear what new experimental or imaging capabilities this technology provides in comparison with existing imaging modalities.

In principle, the bio-flatscope should enable (i) large field of view (FOV), (ii) near cellular resolution, (iii) fast, (iv) 3D imaging in a (v) integrated miniaturized device. The system does achieve large FOV (30x compared to mini scopes, although the comparison should really be

with a conventional epi-fluorescence microscope), but the resolution of the device is poor. The in vivo data presented in mice is mesoscopic imaging. Convincing 3D reconstruction is presented in only hydra at a rather slow speed of 2Hz speed, suggesting there are still limitations to the technology (masking requires high exposure times to discern dim signals?).

Based on the data presented, it will be difficult to apply this technology for 3D imaging in scattering tissue. The system architecture intrinsically allows miniaturization, potentially enabling imaging studies in freely behaving animals performing complex behaviors. However, in its current implementation the device is fairly large and works only in head restrained mice. While I understand a new technology need not check all these boxes, as presented in this manuscript, the in vivo imaging capabilities fall short on several aspects, and this, in my opinion, limits practical applicability and adoption by the broader community in its current iteration.

With regards to my second comment, with the exception of 3D imaging of the hydra, all other imaging demonstrations – in mice as well as in humans can be performed with existing technologies that are as miniaturized as the bio-flatscope with better imaging performance.

In summary, this is a technical improvement in the sense that the lensless flatscope format device can now perform imaging in low light, low contrast conditions, but needs further optimization before it can be practically implemented for in vivo 3D biological imaging.

Below, I describe my major and minor concerns:

We thank the reviewer for recognizing the potential of our work. We believe our responses to the individual points below will clarify the significance of our contributions.

Major Concerns/ questions

Because a few questions from reviewer 1 describe the same concerns. We group them together for clarity:

Category 1, axial resolution:

Depth resolution:

4) What is the depth resolution of the bioflatscope in (i) non-scattering medium (ii) scattering media? Some data needs to be provided perhaps using static 3D phantoms with and without scattering properties, as is commonly done. E.g., See Prevedel* et al Nature Methods Figure 1d-g. This is very important to characterize prior to performing dynamic imaging in vivo.

Benchmarking against existing epifluorescence/lightfield imaging system:

7) At present, this paper lacks a clear and quantified benchmarking of the flatscopes of depth resolution. This could be addressed by perhaps showing (i) comparative 3d imaging in hydra

using a conventional or light field imaging scope, (ii) comparative widefield mesoscopic imaging in mice using a conventional epifluorescence imaging setup.

Lack of in vivo data showing depth scanning:

In my opinion, the primary advantage a flatscope style device is it's the ability to perform 3D imaging. Several questions arise that need clarification.

9) Conventional light field imaging suffers from poor resolution in z direction. What is the depth resolution obtainable with the bio-flatscope?

We thank the reviewer for this suggestion, while some may argue that the large FOV combined with high spatial resolution is the major advantage, we agree that 3D imaging is also an important capability that should be better characterized. We have added new experimental data to characterize the 3D imaging performance including the depth resolution with and without scattering (see **Experiment 1**).

Category 2: 3D imaging in scattering media:

5) 3D reconstruction is shown in Fig.2 in hydra. The data shown in Supplementary Fig. 11 when imaging the mouse brain surface does not appear to show any discernible difference between the different depths. I am currently not convinced with the data provided that 3D dynamic imaging is possible, particularly in scattering tissue.

10) How feasible is 3D imaging in scattering tissue?

17) If image reconstruction was from underneath the surface, how far down could images be reconstructed? What are the limitations to such 3D reconstructions? I think this is important to discuss, and will allow the reader to ascertain the limitations of the technology for in vivo imaging.

We thank the reviewer for bringing this up as it relates to the point above. We do not claim that the bioflatscope can overcome the scattering challenge that faces epifluorescence microscopy. The advantage we have is that we can computational refocus after we capture the data to form focused images of the most superficial regions typically within 200 μm of the surface of the tissue. We have added a new figure (Fig. S14) and a short discussion of this fact in the text :

Line 205-214:

"It should be noted that as implemented here, the Bio-Flatscope does not overcome any of the scattering challenges that face traditional epifluorescence microscopy. Nevertheless, Bio-FlatScope does have the advantage of allowing us to computational refocus after we capture the data. This computational refocusing allows us to form focused images of superficial regions typically within 200 μm of the surface of the tissue [67]. This ability is particularly useful for neural imaging as it can be used to account for motion and curvature of the brain, and is useful for clinical imaging as it expedites the imaging process by moving the focusing procedure backwards, which is favorable to the

patients as well as clinicians. An example of the utility of computational refocusing is shown in our next example: human oral mucosa imaging (Fig. S14)."

In addition we have added a characterization of the axial resolution with and without scattering media (see **Experiment 1**).

Category 3, change of the intensity of the reconstructed image, with respect to working distance and illumination intensity

3) (Part of question 3) Are the slow drifts down in DFF after the "bursts" simply an artifact of the organism drifting in and out of the plane? Or perhaps changing its relative orientation with respect to the bio-flatscope results in changes to the DFF measured?

12) Does image reconstruction result in artifacts? I.e., if a sample with a fixed fluorescence pattern is imaged and reconstructed when located at different distances from the bio-flatscope, how does the intensity vary? The authors provide resolution data. But changes in intensities if any need to be quantified. If the intensity varies, then quantifying DF/F in a moving object such as a hydra would be prone to artifacts arising due to image reconstruction.

18) How does reconstruction algorithm performance change as a function of depth of reconstruction? The supplementary data provided discusses resolution. But for dynamic In vivo imaging, it is important to quantify changes in intensity as well, especially if the end application is to perform imaging in whole moving organisms

We agree with the reviewer that those factors could cause artifacts in our imaging. In order to preclude such possibilities, we added a new simulation (Simulation 1, see above) to characterize the response of our device to different illumination intensity, and another new simulation (Simulation 2, see above) to show that the intensity response of our device does not vary much with depth change over a few hundreds of micrometers, hence drifting in and out of the plane will not cause artifacts.

Here begins our response to the uncategorized questions:

Hardware:

1) The use of the micro prisms coupled to optic fibers is interesting. However, there is no quantification on how well such illumination works. How does illumination vary across the FOV? This is important in the context of dynamic imaging. For instance, if the hydra moves between areas that are dimly lit, as compared to well lit areas, the fluorescence signal is going to vary.

We thank the reviewer for pointing out this consideration. Indeed as the *Hydra* moves to areas of dimmer illumination, the fluorescence intensity would decrease. This is a challenge for all dynamic imaging techniques and is usually overcome by characterizing the illumination

intensity across the FOV and then renormalizing the fluorescence data to account for the variance in illumination. For this renormalization to work we must confirm that the intensity of a reconstructed object varies linearly with the intensity of the illumination.

We added a description of how renormalization can be done in the presence of variations in illumination intensity to supplement material and demonstrate that reconstructed intensity is linear with illumination intensity in Fig. S7.

From the simulation result (Fig S7), we can see that the intensity of each reconstructed bead is linearly proportional to the intensity of illumination applied to it. This will allow us to perform renormalization of fluorescence intensity as is traditionally done for fluorescence microscopy [3].

Description of how illumination is achieved has been added to as **new text** in the Methods - Excitation Light:

“The light was coupled into a multimode fiberoptic cable (Thorlabs M72L01 200 μm diameter core, NA 0.39 or Edmund Optics, #57-749 400 μm diameter core NA 0.22) with an aluminum coated microprism (Edmund Optics 0.18 mm #66-768 or 0.70 mm #66-773) adhered to the exposed tip of the fiber with optical epoxy (Norland, NOA72). The microprism was placed near the surface of the lensless imaging device. The multimode fiber directs the light from the blue excitation LED source, and the aluminum coated microprism works as a mirror to redirect the light emitting from the fiber to the surface of the mouse brain (see Methods). As a result, the final light spot is similar to the typical output pattern of a LED source.”

Insufficiency of presented data:

2) Mice experiments: The reporting summary states $n = 3$ animals, which seems like the bare minimum. There are several factors to consider in the design of these experiments - efficacy of viral injections, variation in expression levels of GCaMP, variation in GCaMP activity from mouse to mouse etc, which necessitates higher number of animals. Further, Figure 3 and 4 show representative data from a single mouse and statistical analysis is performed on the data from the single mouse.

How does the performance of the bio-flatscope vary when taking into consideration all the biological variables?

- Can authors provide reconstructions from multiple animals?
- What is the average signal to noise ratio obtained in multiple mice?

To minimize the number of animals used for these experiments as advised by our IACUC we focused here on a proof of concept demonstration that involved imaging in 7 mice. Total number of mice imaged in the experiments is seven and reconstructions can be successfully made with them. We selected two in the result session based on the expression level (shown in Fig. 3(S8) and Fig. 4(S13)), surgery performance and mouse activity. Similar selection criteria are used in other imaging experiments and do not represent a limitation of Bio-FlatScope.

Sample sizes of three mice are common for other proof-of-principle imaging technology papers: Ghosh et al. Nature Methods, 2014 (n=3 mice), Prevedel et al. Nature Methods, 2014, (n=1 zebrafish).

We have added text describing the selection criteria and outcome of all 7 mice preparations and 3 imaging experiments:

Methods - Animals for in vivo mouse brain imaging

“Wild-type C57BL/6 mice (n=7) from Charles River Laboratories were used for this study. Animals are housed with standard 12 h light/dark cycle with ad libitum food and water. All animals (n=7) were injected with the adeno-associated viral vector AAV9.CamKII.GCaMP6f.WPRE.SV40 (Penn Vector Core). Total number of mice imaged in the experiments is seven and reconstructions can be successfully made with them. We selected two in the result session based on the expression level, surgery performance and mouse activity.”

3) Hydra imaging: Calcium imaging data is provided from just one hydra vulgaris experiment.

The single *Hydra* experiment provided enough data for us to confirm proof-of-concept imaging capabilities. These data have been moved to supplemental. We do not use these data to make claims about performance since this imaging has been de-emphasized in the revised draft.

For the non-hydra experts, it is difficult to make sense of the data provided. Are the Ca signals shown typical for certain types of 3D body shapes of the organism? Are they consistent across multiple hydra imaged? What unique insight about hydra behavior or activity as imaged by calcium imaging is being revealed by the bio-flatscope? With the representative data provided it is currently not possible answer these questions.

Hydra data was intended to be used to show 3D imaging using a model organism with known calcium dynamics - namely calcium bursts that correspond to contractions (Badhiwala et al., 2018). 3D imaging has now been demonstrated using 3D bead samples and immobilized *Hydra* (**Experiment 1**). Furthermore the mobile *Hydra* imaging has been de-emphasized and moved from the main text to supplemental information.

How does this compare with imaging performed using conventional instruments? Without a proper benchmark, e.g., with a conventional light-field imaging system, it is difficult to ascertain how well 3D imaging really work. While 3D imaging of freely floating hydra may be difficult to perform using conventional imaging setup, can an experiment be performed with restrained hydra that allows a comparative study?

We agree with the reviewer that further comparing our *Hydra* imaging results with a ground truth will help to strengthen our demonstration of 3D imaging capability. We have added imaging results of immobilized *Hydra* to the manuscript as Fig. 2g and also included in this response document as **Experiment 1**.

6) “Here the large FOV and 3D reconstructions help us capture movies of whole animal calcium dynamics, which can be used by neuroscientists to study information processing in these small model organisms” Expanding on this point may help the reader ascertain what kind of information processing one may perform. At present, the resolution seems to be too limited to image single cells, or even mesoscale flow of calcium activity within the organism.

Imaging mesoscale activity in small model organisms is very useful for monitoring activity and studying information processing. For imaging neural activity in *Hydra*, researchers often focus on a group of neurons in the peduncle (foot region) with GCaMP because they are known to fire together. To monitor the muscle activity in *Hydra*, the whole body GCaMP activity is recorded [4]. Recording from either a large group of cells or even from the whole animal with distinct fluorescent levels when the cells are active/inactive is very useful for studying the small model organisms. In Figure S4, we demonstrated the ability of recording the whole-animal dynamic calcium activity in *Hydra*, and the results indicate strong calcium responses during contraction events as reported previously [4].

8) From the data provided, it looks like the *Hydra Vulgaris* imaging was done in 3D, while imaging in mice and humans was performed in 2D. While supplementary data shows image reconstructions at different depths from the sensor, the different reconstructions appear mostly similar -whereas, typically one should be able to trace 3D blood vessel geometry with even low magnification widefield objectives. Are there any differences in the mesoscale activity maps from different depths?

We thank the reviewer for carefully reviewing these images and expressing concerns. We have added a new Fig. S14 which shows that we can computationally refocus to different depths for resolving the microvasculature. Regarding “2D” vs “3D” capture, our lensless system always captures the entire volume of light from the region of interest. This means we always have the capability to reconstruct 3D information. In practice however, certain samples (like the imaging in mouse and human) have limited signal available from farther depths (as with traditional lensed microscopy) rendering epifluorescence reconstruction at those depths highly limited due to tissue scattering.

Without scattering we can more fully characterize the 3D imaging capabilities of the Bio-Flatscope compared to confocal imaging (see **Experiment 1**)

11) How does resolution and image intensity vary as a function of depth of reconstruction?

We thank the reviewer for bringing these two important performance metrics. To respond, we added **a new simulation (Simulation 1, see above)** to characterize the response of our device to different illumination intensity. As Bio-FlatScope reported in this manuscript was designed to work at a relatively narrow working distance range (~1 mm), we did not characterize its lateral resolution in a large depth range. The lateral resolution acquired by analysing USAF target reconstructions remained consistent within the ~1 mm depth range. As a supplementary information to this response, we provided a lateral resolution characterization of the phase mask used in Bio-FlatScope over ~200 mm depth range in another paper (Fig. 11) published

last year: V. Boominathan, J. K. Adams, J. T. Robinson, and A. Veeraraghavan, “PhlatCam: Designed Phase-Mask Based Thin Lensless Camera,” *IEEE Trans. Pattern Anal. Mach. Intell.*, vol. 42, no. 7, pp. 1618–1629, 2020.

Figure 1 in the manuscript provides data from planar samples. Some data needs to be provided perhaps using static 3D phantoms with and without scattering properties, as is commonly done. E.g., See Prevedel* et al *Nature Methods* Figure 1d-g. This is very important to characterize prior to performing dynamic imaging in vivo.

We thank the reviewer for this suggestion. Additional simulation and experiment with fluorescent beads in non-scattering and scattering media to characterize lateral and axial resolution have been added at the beginning of this document as **Experiment 1**.

In vivo mouse imaging:

13) Calcium traces are currently shown over very long lengths scale. For clarity, zoomed in view of a few seconds of data should be provided for both flatscope and epi-fluorescence data. Without that data, it is difficult to qualitatively and quantitatively compare the data shown in Figure. 3.

We thank the reviewer for this suggestion. We have provided the figure (Figure S17) showing a zoomed in views of $\Delta F/F$ near the stimulation events I and II for comparison of Bio-FlatScope and epifluorescence. Note that these measurements were not taken simultaneously, so we do not expect an exactly identical response.

Fig. S17. Zoom-ins of Fig 3D&E. These zoom-ins show $\Delta F/F$ corresponding to the stimulation events for comparison of Bio-FlatScope and epifluorescence. Note that these measurements were not taken simultaneously, so we do not expect an exactly identical response.

14) The clusters shown in Fig. 5 are very large - >100 micrometers in some cases. This will have several cells within, with a few cell's dominating the responses measured. So the claim of 'near cellular resolution' is a stretch.

We thank the review for pointing out this potential confusion. We have clarified that "near cell resolution" refers to the fact that some clusters are roughly 10-20 μm in diameter. We have clarified the text to be explicit about the resolution of the functional clusters:

Line 186-188:

"The calcium activity in the smallest of these ROIs (10-20 μm diameter) show unique temporal dynamics suggesting that calcium dynamics might be recovered with near cellular resolution."

Methods:

15) In contrast with traditional single photon imaging, data here is reconstructed. There is little detail on how the reconstructed images were analyzed. It is possible the analysis methodology proceeds along similar lines to say data obtained from mini scopes, but for the sake of completeness, this information should be provided. For instance, I had several questions for which I did not get an answer in the methods section.

We thank the reviewer for asking more technical details, which we have updated as described in our response below to each of these questions.

16) Since images can be reconstructed at multiple planes, what plane was chosen for reconstruction? The surface of the brain or a plane slightly underneath?

The plane for reconstruction was chosen by selecting the depth at which blood vessels are reconstructed with the best sharpness in the region of interest. We have updated the manuscript:

Methods - Reconstruction:

"In our experiments, the plane for reconstruction was chosen by selecting the depth at which features of interest (e.g. blood vessels) are reconstructed with the best sharpness in the region of interest."

- Currently, the reconstructions are shown as distance from the sensor, which is not informative.

We have clarified that this working distance is the distance between the outermost surface of the Bio-Flatscope (the interference filter) and the imaging plane (similar to a microscope working distance), and added it as new text in the revised manuscript:

Methods - Calibration:

“This “distance”, which is mentioned many times throughout the text, is defined as the distance between the outermost surface of the Bio-Flatscope (the interference filter) and the imaging plane, which is similar to a microscope working distance.”

-How is data analyzed once a reconstructed image is obtained?

Once a reconstructed image is obtained we treat it as an epifluorescence image for analysis. We have added the following to the text: “Once a 2D image is reconstructed it can be analyzed as a conventional epifluorescence image.”

-What does DF/F signify in Fig. 2C? Is this the change in DF/F over the whole FOV? Is this the change in DF/F over the whole FOV averaged across multiple z-planes after reconstruction. Or is it simply DF/F for the whole raw image?

It is the whole FOV at a single z-plane after reconstruction.

Minor concerns

1) There is a lack of detail on the hardware of the bio flatscope. More information will be helpful.

(a) What is the function of the interference filter?

The function of the interference filter is to reject the majority of blue excitation light scattered and reflected from the sample to the sensor.

(b) The use of the absorptive filter is interesting. How well does it work? Is there a particular need to use an absorptive filter as against a regular glass filter? For the sake of completeness, this information should be provided.

It works well (OD3) to reject the remaining off-angle excitation light that does not get rejected well by the interference filter.

We have included the information for minor concerns 1) a) and b) in Methods - Illumination strategy for fluorescence imaging in the revised manuscript.

2) Since the data shown in mice is changes in fluorescence in bulk tissue. It will be prone to artifacts arising from hemodynamic changes. This should be acknowledged.

We agree this is an important point to communicate to the reader we have added:

Line 136-137:

“Hemodynamic changes were not considered in our reconstruction and data analysis”

3) In the supplementary table, the authors compare the performance of the bio-flatscope to other miniaturized imaging devices. In its current form, the bio-flatscope is still too bulky and all

the experiments performed by the authors are on head fixed animals. The comparison need to be made with respect to a conventional wide field epi-fluorescence scope or a light-field imaging setup.

The current weight of the bioflatscope is approximately 5g and with an additional miniaturization of the electronic components can bring that below 3g. Because miniature microscopy is the main use case of bio-Flatscope, we have already included a comparison table for other imaging systems that weigh 5g or less in our original manuscript (**Table S1**).

4) Scale bar would be helpful in Figures 1a and 3a.

We thank the reviewer for this suggestion and have added them accordingly.

Reviewer #2:

The article “In vivo imaging with a flat, lensless microscope” by Adams et al. presents an interesting and exciting extension of the group’s previous work on the FlatScope (Ref 10. in the manuscript) to live, fluorescent labeled and unlabeled specimens for the first time in a low-profile, compact lens-less microscope. They utilized a specialized phase mask optimized for dense, low contrast fluorescent samples. Several exciting applications were demonstrated such as on fluorescent test samples (USAF and mouse brain tissue), live samples such as the *Hydra vulgaris* (with some limited 3D resolution), and perhaps most exciting with live mounted imaging of mice brains during locomotion and of human oral mucosa imaging. While the work is valid and exciting, in the current instantiation it appears more incremental and not of as high impact as other articles in Nature Biomedical Engineering.

Therefore, at the current time I cannot recommend publication in the journal without addressing the following concerns and issues.

In the opinion of this reviewer, the most novel and interesting aspects of the current manuscript

are the application of the FlatScope (Ref. 10 in the manuscript) to live fluorescent labeled tissues and human mucosa. This is apparently a first for lens-less imaging technologies that have multiple advantages such as the small form factor, light weight, and less invasive live biological imaging as demonstrated with the *in vivo* calcium imaging in mice brains and human mucosal region imaging. Surely the human subjects would feel more comfortable and less worried to have a small chip akin to something like a modern dental x-ray detector put near their mouth than a bulky optical microscope. However, the impact and the novelty of the work is brought into question by numerous other demonstrations of live mouse brain imaging while allowing animal locomotion such as the work by Ziv et al. Nature Neuroscience 16, 264 (2013).

Furthermore, other groups have demonstrated low profile, len-less, even single shot fluorescent imaging cameras such as Antipa et al. Optica 5, 2334-2536 (2018).

Furthermore, the demonstrated single-shot 3D imaging of the Hydra, has comparable but less convincing results as compared to other published results such as Yanny et al, Light Sci Appl 9, 171 (2020), although this group did not do *in vivo* imaging.

We thank the reviewer for encouraging us to better place our work in context. While many of the examples above perform better than our device in one or two metrics, there is no lensless microscopy demonstration *in vivo* that simultaneously achieves a large FOV and sub -10 μm spatial resolution. To help make this point we updated **Table S1** and added a new **Table S2** to compare different miniature microscopes and lensless imaging techniques..

Table S1 shows that Bio-FlatScope has the largest field of view compared to any previously-reported head-mounted microscope with sub-10-micron resolution. (including Yanny et al. (Miniscope3D), and Ziv et al.)

Table S2 compares Bio-FlatScope to other lens-less imaging techniques. In this table we can see that only Bio-FlatScope demonstrates the ability to image at sub-10-micron resolution in scattering mammalian tissue.

Table S1. Comparison of Bio-FlatScope to miniaturized, head-mounted microscopes.

Microscope	Lateral resolution	Field of view
Bio-FlatScope ^a	~8.8 μm	~16.2 mm^2
Integrated miniature microscope [11] ^b	~2.5 μm	0.48 mm^2
Fiber bundle microscope [12]	~4.9 μm	~0.3 mm^2

Two-photon miniaturized microscope [13]	0.64 μm	$\sim 0.03 \text{ mm}^2$
Miniaturized light-field microscope [14]	6 μm	$\sim 0.42 \text{ mm}^2$
Miniscope3D [15]	2.76 μm	$0.9 \times 0.7 \times 0.39 \text{ mm}^3$
Mini-mScope [16]	39.36 ~ 55.68 μm	80 mm^2

^a Note that for Bio-FlatScope to reach the same < 3g weight as head-mounted microscopes, the sensor PCB will need to be miniaturized like the miniature lens-based microscopes shown here.

^b The imaging device used in Ziv et al. is originally reported in this work.

Table S2. Comparison of Bio-FlatScope to state-of-the-art lensless imaging technologies.

Microscope	Any in vivo imaging	in vivo imaging of mammalian tissue	Working distance ^a	Lateral resolution	Axial resolution	Field of view
Bio-FlatScope	Yes	Yes	$\sim 4 \text{ mm}$	$\sim 8.8 \mu\text{m}$	50 μm	$\sim 4 \times 4 \text{ mm}^2$
FlatScope [17]	No	No	200 μm	7 μm	15 μm	$2.5 \times 2.5 \text{ mm}^2$
On-chip microscope [18]	Yes	No	1.5-3 mm	8 μm	50 μm	$2 \times 1.5 \text{ mm}^2$ ^b
DiffuserCam [19] ^{c1}	No	No	20 mm	45 μm	336 μm	$36 \times 20 \text{ mm}^2$ ^{c2}
CM ² [20]	No	No	12 mm	7 μm	200 μm	$8 \times 7 \text{ mm}^2$

^a The working distance is defined as the distance at which the parameters in the table are characterized.

^b The authors did not explicitly mention their FOV in the manuscript. From their fluorescence bead imaging results shown in Fig 8 we estimated the FOV to be $2 \times 1.5 \text{ mm}^2$.

^{c1} This work reports a camera with 45 micron spatial resolution and long working distance rather than a microscope as described here. Nevertheless, Figure 1 of our manuscript compares our Bio-FlatScope to a diffuser-based device.

^{c2} The FOV of this work is reported as angle instead of length, which is typical for cameras. The authors report their FOV as half-angle 42° in x and 30.5° in y, corresponding to 36 mm in x and 23.6 mm in y at 20 mm working distance.

Furthermore, in my opinion, the demonstration of the 3D imaging of the hydra is not convincing and appears more like a 3D extrusion of a 2D image. However, it is difficult to determine this with the current images. More images and analysis, especially along the third dimension would allow more careful and accurate assessment.

We added new experiments to characterize the 3D imaging ability of Bio-FlatScope and demonstrate improved single-shot 3D reconstruction of *Hydra* (see **Experiment 1** and revised Figure 2).

Finally, this leads me to an additional final concern about the manuscript in that the authors attempt to show so many example applications (each of which is exciting and valid), but it reduces the amount of information that can be placed in the main body and dilutes the impact of each of the applications. I would recommend on focusing on the two most transformational and impactful applications for Nature Biomedical Engineering which in my opinion are the live mouse brain imaging and human oral mucosa imaging. These applications most clearly highlight the potential clinical applications due to its compactness, light weight, and lens-less nature. Below I list other technical items that should be addressed:

We thank the reviewer for this suggestion and have restructured our manuscript by moving *Hydra* section from the main text to supplemental information, and added a new figure (new Fig. 2) for characterization.

Major technical criticisms or questions:

1) The current manuscript is extremely dense and spends too little time on too many applications. I would recommend moving some of the information to the SI (or possibly a separate article) such as the 3D demonstration with the *hydra vulgaris*. To me, this demonstration does not support the main tenants of the paper and feels like a distraction. This on top of the fact that the claim to 3D resolution seems weak with the current small sub-figures in Figure 2a. It is difficult for me to access the 3D image quality or resolution in the third dimension from the current Fig 2, especially when comparing the top and bottom images. Other information that could be moved to the supplemental information is the description of the mouse brain phantom (it is not even discussed in the main body but in the SI now), and perhaps Fig 3g as I am not sure exactly what it depicts. Furthermore, what resolution is claimed in the 3rd dimension and how is this possible with a relatively flat phase mask compared to for example Antipa et al. *Optica* 5, 2334-2536 (2018) work.

We thank the reviewer for this suggestion and have restructured our manuscript. To better demonstrate the 3D imaging capability of Bio-FlatScope, we added **a new experiment (Experiment 1, see above)** to show the axial resolution of our device.

2) There is not enough information currently to evaluate the phase mask used. What are the spatial frequencies encoded in the phase mask and the larger range focus characteristics (beyond the few mm range shown in Fig S10 and S11).

We thank the reviewer for bringing these two important performance metrics. Please see panel B of Fig. 1 and Fig. S1 of our manuscript for the spatial frequency encoded in our phase mask. As Bio-FlatScope reported in this manuscript was designed to work at a relatively narrow working distance range (~1 mm), we did not characterize its lateral resolution outside of this design range. Other work using similar phase masks have focused on photography applications with long working distances e.g. Fig. 11 from V. Boominathan, J. K. Adams, J. T. Robinson, and A. Veeraraghavan, "PhlatCam: Designed Phase-Mask Based Thin Lensless Camera," IEEE Trans. Pattern Anal. Mach. Intell., vol. 42, no. 7, pp. 1618–1629, 2020.

A detailed description of phase mask evaluation was added as the Supplementary Information Section 1:

"In lensless imaging, the scene is encoded onto the sensor by convolution of the scene with a PSF. From convolution theorem, we can infer that for maximal information transfer, large and flat magnitude spectrum is desirable in the PSF. The deconvolution of PSF involves the inversion of the PSF's frequency spectrum, and low values of the magnitude spectrum can lead to amplification of noise.

We compared the Modulation Transfer Function (MTF) of our proposed contour PSF with PSFs designed for other lensless imaging systems (Fig. S1). The MTF is computed as the radially averaged magnitude spectrum of the PSFs. The magnitude spectrum of our proposed contour PSF remains large for the entire frequency range, which indicates better invertibility characteristics."

Minor technical criticisms or questions:

1) How is the working distance or sample to BioFlatScope distance adjusted? Additional figures or description of this, at least in the SI would be helpful. What is the effective depth of focus?

The working distance is adjusted to optimize the imaging performance and to match the designed optimal working distance of the phase mask, for this specific Bio-FlatScope prototype is ~4mm.

The Depth of Focus, for Bio-FlatScope is characterized to be the range in which the PSF maintains a sufficiently good quality, which for this specific prototype is 2-7mm from the surface of the interference filter. Please note that we can always find depth-matched PSFs to reconstruct the image.

2) How sensitive is this working distance to the reconstruction algorithm?

The working distance is irrelevant to the reconstruction algorithm. It only depends on the PSF of the phase mask, which can be designed for a specific distance range.

3) How computationally intensive is the reconstruction algorithm (single processor/multiple processor, compute time, array size of data, etc)? These details would help the reader evaluate the computational cost of, for example, the movies in the supplemental information.

We haven't done any speed optimization on our reconstruction code. The current run time on a 8-Core Processor (AMD Ryzen 7 3700X, 3.59GHz) is ~1.3s for a frame contains 3072*2048 pixels using MATLAB. Each frame is stored as a 16-bit TIFF file.

4) Fig. 2 and Page 5, what volume of fluid is the *hydra vulgaris* polyp confined to? Additional details about this experiment would be helpful as it is obvious from the movie that the sample is tumbling freely, but it is not clear why it leaves the image and then returns. Does this point to some limited 'depth of focus'?

We thank the reviewer for this concern. We have added the following text to Methods -*Hydra vulgaris* to help clarify that it does not indicate any DOF limitation.

"The Hydra was provided sufficient media to move around freely, in this case ~3-5 mm of depth within the petri dish (described in Methods). The movie shows reconstruction video from a single depth plane hence witnessing the Hydra "leaving the image and returning". This has been clarified in the body of the text as well as the captions."

Please also refer to our response to minor questions 1) for DOF of Bio-FlatScope explanation:

"The Depth of Focus, for Bio-FlatScope is characterized to be the range in which the PSF maintains a sufficiently good quality, which for this specific prototype is 2-7mm from the surface of the interference filter. Please note that we can always find depth-matched PSFs to reconstruct the image."

5) In Fig 5b, there are a number of vertical and horizontal artifacts in the oral mucosa images as compared to the 4x objective microscope images. We assume these are from computational image reconstruction algorithm, but they should be addressed and discussed. Is there a way to mitigate or remove them? Can they not be addressed with appropriate background subtraction or white field removal?

We thank the reviewer for bringing this up. We hypothesize that these artefacts are due to extraneous light reaching the sensor from outside the FOV and could be mitigated by a tighter illumination control.

6) In the methods section, line 291-293, a description of the range of depth over which the calibration is performed is given, but no distances are mentioned. Please mention those distances.

We thank the reviewer for this suggestion and have added the following text to Methods - Calibration:

“Images of the PSF were captured for each depth (every 20 μm) over a distance range from 2 mm to 7 mm. The bead is approximately aligned to the center of the phase mask.”

7) The algorithm described on lines 308-311 for reconstruction of 3D volumes seems to treat each layer independently. This seems like a poor assumption for dense tissue where multiple scattering effects could be important from subsequent layers. Is there a way to address this or is it not important for true 3D imaging in tissues from lens-less imaging modalities?

We thank the reviewer for this question and have added the following text to Methods - Reconstruction as a response:

“3D reconstruction from a single 2D capture is still an under-determined problem, and compressed sensing theory provides some guidance on samples that can be reliably reconstructed [7-9].”

8) Additional text describing how the MTF and resolution is determined/measured in the supplemental Fig S1 and S5 would be helpful. The figure captions are not sufficient.

The MTF is calculated with simulated PSFs under the definition of MTF. We improved our description of the phase mask and MTF calculation in section 1 of the supplementary information. The curves in Fig. S5 shows the intensity change across the bars in the USAF target reconstruction.

9) Moving Fig. S3 earlier in the draft, perhaps near figure 1, would help the reader to understand the exact geometry of the illumination.

We thank the reviewer for this suggestion.

References:

1. F. Ye, B. W. Avants, A. Veeraraghavan, and J. T. Robinson, "Integrated light-sheet illumination using metallic slit microlenses," *Opt. Express* 26, 27326–27338 (2018).
2. D. Miyamoto and M. Murayama, "The fiber-optic imaging and manipulation of neural activity during animal behavior," *Neurosci Res* 103, 1–9 (2016).
3. Nguyen, J. P. et al. Whole-brain calcium imaging with cellular resolution in freely behaving *Caenorhabditis elegans*. *Proc. Natl. Acad. Sci. U. S. A.* 113, E1074–E1081 (2016).
4. J. R. Szymanski and R. Yuste, "Mapping the whole-body muscle activity of *Hydra vulgaris*," *Curr Biol* (2019).
5. Min, E. *et al.* Measurement of multispectral scattering properties in mouse brain tissue. *Biomedical Optics Express* 8, 1763 (2017).
6. Hoshi, Y. *et al.* In situ estimation of optical properties of rat and monkey brains using femtosecond time-resolved measurements. *Scientific Reports* 9, (2019).

7. Michalet, X. & Weiss, S. Using photon statistics to boost microscopy resolution. *Proceedings of the National Academy of Sciences of the United States of America* **103**, 4797–4798 (2006).
8. D. L. Donoho, "Compressed sensing," *IEEE Trans. Inf. Theory* 52(4), 1289–1306 (2006).
9. E. J. Candès and M. B. Wakin, "An introduction to compressive sampling," *IEEE Signal Process. Mag.* 25(2), 21–30(2008).
10. E. J. Candès, "The restricted isometry property and its implications for compressed sensing," *CR Math.* 346(9-10),589–592 (2008).
11. K. K. Ghosh, L. D. Burns, E. D. Cocker, A. Nimmerjahn, Y. Ziv, A. E. Gamal, and M. J. Schnitzer, "Miniaturized integration of a fluorescence microscope," *Nat Methods* 8, 871–878 (2011).
12. M. Kim, J. Hong, J. Kim, and H. Shin, "Fiber bundle-based integrated platform for wide-field fluorescence imaging and patterned optical stimulation for modulation of vasoconstriction in the deep brain of a living animal," *Biomed. Opt. Express* 8, 2781 (2017).
13. W. Zong, R. Wu, M. Li, Y. Hu, Y. Li, J. Li, H. Rong, H. Wu, Y. Xu, Y. Lu, H. Jia, M. Fan, Z. Zhou, Y. Zhang, A. Wang, L. Chen, and H. Cheng, "Fast high-resolution miniature two-photon microscopy for brain imaging in freely behaving mice," *Nat Methods* 14, 713–719 (2017).
14. O. Skocek, T. Nöbauer, L. Weilguny, F. Martı́nez Traub, C. N. Xia, M. I. Molodtsov, A. Grama, M. Yamagata, D. Aharoni, D. D. Cox, P. Golshani, and A. Vaziri, "High-speed volumetric imaging of neuronal activity in freely moving rodents," *Nat. Methods* 15, 1–4 (2018).
15. K. Yanny et al., "Miniscope3D: optimized single-shot miniature 3D fluorescence microscopy," *Light Sci. Appl.*, vol. 9, no. 1, 2020
16. Rynes, M. L. et al. Miniaturized head-mounted microscope for whole-cortex mesoscale imaging in freely behaving mice. *Nat. Methods* 18, 417–425 (2021).
17. J. K. Adams, V. Boominathan, B. W. Avants, D. G. Vercosa, F. Ye, R. G. Baraniuk, J. T. Robinson, and A. Veeraraghavan, "Single-frame 3D fluorescence microscopy with ultraminiature lensless FlatScope," *Sci. Adv.* 3, e1701548 (2017).
18. Grace Kuo et al. "On-chip fluorescence microscopy with a random microlens diffuser," *Opt. Express* 28, 8384–8399 (2020)
19. N. Antipa, G. Kuo, R. Heckel, B. Mildenhall, E. Bostan, R. Ng, and L. Waller, "DiffuserCam: lensless single-exposure 3D imaging," *Optica* 5, 1 (2018).
20. Y. Xue, I. G. Davison, D. A. Boas, and L. Tian, "3D Fluorescence Imaging with a Computational Mesoscope," *Sci. Adv.*, no. October, pp. 1–16, 2020.

Rebuttal 2

Editorial comments:

Dear Dr Robinson,

Thank you for your revised manuscript, "In vivo imaging with a flat, lensless microscope", which has been seen by the original reviewers and one more expert. In their reports, which you will find at the end of this message, you will see that the reviewers acknowledge the improvements to the work and raise a few additional technical criticisms that we hope you will be able to address. In particular, we would expect that the next version of the manuscript provides:

- * improved characterisation of the Bio-FlatScope's in vivo performance, including axial resolution and illumination parameters;
- * improved technical accuracy of the in vivo image resolution claimed, please avoid using ambiguous language (such as "approaching cellular resolution"), also expanding the description of the RPCA component of the study; please consider creating a table, for the data shown in Fig 4, describing, for each ROI, its size and its diameter (pixels and μm).
- * please consider contextualising the in vivo image performance claims to include parameters that define digital image resolution, such as the Nyquist limit, dynamic range, and SNR; please consider referring to the in vivo resolution as the "post-processing" digital image resolution, clarifying that the value describes specifically the mice/samples imaged in this study.
- * the in vivo image resolution could be further discussed in a limitations paragraph within the discussion where signal intensity, autofluorescence and filters can be used to list criteria that could be optimized to improve the performance of the Bio-FlatScope.

We thank the editor for gathering these reviews and the opportunity to improve this manuscript by addressing them. Below we summarize the major changes, provide a list of new analysis, followed by a point-by-point response to the reviewers:

Summary of Major Changes:

1. **Additional *in vivo* imaging performance analysis:** We added new analysis using FWHM to characterize our computational refocusing capability in human oral mucosa (**Analysis 1**, Figure S14). We also expand our description on functional brain imaging by adding new analysis to show the number of pixels and areas of each clustered ROI in mouse brain imaging shown in Figure 4 (**Analysis 2**, Table S3). Finally, we improved our description by removing the ambiguous language related to "cellular resolution", adding more description on RPCA, and adding discussion on limitations we faced during *in vivo* imaging.

List of additional experiments, simulations and analysis:

Analysis:

Analysis 1 (Reviewer 1)

(included in the revised manuscript as Fig. S14)

To demonstrate the computational refocusing ability in mammalian tissue, we performed FWHM analysis over a line drawn across the small vessel in human oral mucosa shown in the blue square in Fig. S14a&b. The width of the vessel expands more than two times within a $\pm 120 \mu\text{m}$ range.

Fig. S14. Computational refocusing in human oral mucosa. (A) Bio-FlatScope reconstructions at 2.88mm depth. The image is out of focus and some of the small features cannot be reconstructed. (B) Bio-FlatScope reconstructions at 3.04mm depth. The image becomes sharp and small vessels can be reconstructed at high contrast. Scale bar, 100 μm . (C) FWHM of the intensity across the small vessel in the blue square shown in panel A and B at different imaging depths.

Analysis 2 (Reviewer 1)

(included in the revised manuscript as Table. S3)

To quantitatively demonstrate our *in vivo* performance on mouse brain imaging, we included a new supplementary **Table S3** which contains the number of pixels and areas of each clustered ROI in mouse brain imaging shown in Figure 4. We hope this would help clarify and acknowledge the limitations faced in functional brain imaging.

Table S3. Number of pixels and areas of each clustered ROI in Figure 4

Cluster number	Number of pixels	Area (mm^2)
1	316	0.0052
2	144	0.0024

3	88	0.0014
4	300	0.0049
5	316	0.0052
6	308	0.0050
7	208	0.0034
8	288	0.0047
9	444	0.0072
10	700	0.0114

Materials added and edited:

Figures and tables:

1. Added Fig. S18 and Fig. S19.
2. Added Table S3.
3. Updated Fig. S14

Text:

1. Text edits are marked in the revised manuscript as a different color.
2. Removed all the ambiguous descriptions related to “cellular resolution” in the manuscript.

Reviewer #1:

The revised manuscript has addressed most of my comments from earlier. But there are a few remaining concerns. My major concern is still the poor quality data in vivo mouse imaging data presented in the manuscript and the unsubstantiated claims the authors make based on this.

Major concerns

Axial resolution and focusing during in vivo functional imaging: A lot of paper describes 3D refocusing. Based on the data now shown in the revised Figure 2, the axial resolution in scattering medium is 80 micrometers. This is in test samples and is rather poor. The data

shown in the in vivo functional imaging will have further confounding factors that will make the axial resolution worse. Indeed the qualitative data shown in Figure S13 shows no discernible change in sharpness of the image captured over 300 micrometers. There is some data shown – Can the authors please provide an analysis FWHM of the intensity over a line drawn across any of the major blood vessels imaged in the montage S13A? I suspect there will be very little change in FWHM measurements. This will provide a true qualitative indication of the actual focusing capabilities during in vivo functional imaging experiments. This limitation really needs to be acknowledged and discussed in the main manuscript. The current claim of resolution really is in test, fixed samples, and this should be distinguished from the actual resolution obtained in functional imaging experiments.

We thank the reviewer for suggesting we make this important clarification. As suggested, we have updated the **Figure S14** to provide the FWHM analysis and demonstrate the computational refocusing capability. We performed FWHM analysis over a line drawn across the small vessel shown in the blue square in Fig. S14 a and b. The width of the vessel expands more than two times within a $\pm 120 \mu\text{m}$ range, which we believe more clearly demonstrates our computational refocusing capability in an in vivo imaging experiment (see **Analysis 1**).

We agree that the claimed resolution is in test, fixed samples, and this is different from actual resolution obtained in brain functional imaging. We included a new supplementary **Table S3** which includes the number of pixels and areas of each clustered ROI in functional mouse brain imaging shown in Figure 4 (see **Analysis 2**). We hope this would help clarify and acknowledge the limitations faced in functional brain imaging. We also added the discussion of this limitation in *Discussion*:

Lines 247-251:

"While the resolution shown in fixed samples (Fig. 1&2) appears to meet the needs for microvascular imaging and Ca^{2+} imaging, the *in vivo* resolution could become worse, as the scenes are usually dense, dim, low-contrast, and with increased complexity [10,11]. Although these challenges have already been greatly overcome by Bio-FlatScope, there is still room for improvement."

"This ability is particularly useful for neural imaging as it can be used to account for motion and curvature of the brain". Correction for motion, particularly when the axial resolution is poor is highly unlikely and non trivial. This is a speculative sentence and not substantiated by any data provided. This part of the sentence needs to be removed.

We apologize for this confusion caused by our wording "account for motion". We did not mean that we were able to correct motion blur caused by the animal movement. What we meant here is that with the computational refocusing ability of Bio-FlatScope, the axial position change of the brain surface can be computationally accounted for by selecting PSFs at different depths during reconstruction. As a comparison, such refocusing is done by mechanically adjusting the lens assembly in conventional lens-based imaging systems, where a real-time automatic focusing system is necessary. Shifting this refocusing process from capturing step to

post-processing step could expedite the imaging process, which is favorable to patients as well as clinicians.

We updated the description of the application of computational refocusing in section “*In vivo epifluorescence calcium imaging in mice*”:

Lines 271-277:

"This ability is particularly useful for neural imaging as this post-processing step can be used to account for axial position change of the brain surface and curvature of the brain. With the computational refocusing ability of Bio-FlatScope, the axial position change of the brain surface can be accounted for in computational post-processing by selecting PSFs at different depths during reconstruction. The key advantage of refocusing during post processing is that it does not require the additional hardware and real-time systems needed to perform the physical manipulation of the optics that is necessary for autofocusing during an experiment."

Functional mouse imaging data: The revised manuscript has still not addressed the issue of benchmarking with respect to a conventional scope as I had commented in my previous review. Given the poor axial resolution in scattering medium, a comparison can simply be made between the flat scope and the epifluorescence imaging shown in D.

– For the clusters that are identified, can the authors please provide spontaneous DFF traces for both the data captured using the epifluorescence microscope and the flatscope? This will provide the readers with a better comparison between the two modalities and help benchmark the data.

We added a new supplementary **Figure S18** to show the DF/F traces for data captured using epifluorescence microscopy with a 4× objective in Figure 4d. The epifluorescence data was not captured simultaneously with the Bio-FlatScope data so the traces are different compared to Figure 4, but qualitatively similar. Due to the small form factor and close working distance of Bio-FlatScope, it would be extremely difficult to achieve simultaneous capture with both devices.

Fig. S18. Extracting Ca^{2+} signals from epifluorescence recording with a 4x objective. (A) Cropped region of a single frame with a high-activity region shown by a dashed box. Scale bar, 100 μm . (B), Zoom-in on the region of high-activity with overlay of clusters determined (from Bio-FlatScope data) through post-processing using RPCA and k-means. Scale bar, 50 μm . (C) $\Delta F/F$ traces from the video over during two-minute recording corresponding to the clusters. (D) Maximum projection from epifluorescence recording of same high activity region with overlay of clusters. Scale bar 50 μm . (E) Maximum intensity projection of epifluorescence recording data processed using RPCA with overlay of clusters. Scale bar 50 μm . (F) ΔF traces from epifluorescence recording after post-processing using RPCA. Note that the clusters shown here were found using the Bio-FlatScope data.

“The calcium activity in the smallest of these ROIs (10-20 μm diameter) show unique temporal dynamics suggesting that calcium dynamics might be recovered with near cellular resolution.” A number of issues with this.

- None of the clusters that the authors show traces for are anywhere close to 10-20 micrometers. The smallest cluster highlighted in the traces are 6 and 3. Both appear to be ~30-40 micrometers in diameter. The trace corresponding to 3 is noisy in both Fig. 4c and Fig. 4f.
- The ‘cell like activity claim’ hinges on this figure. But there is no reference to data presented in Fig. 4C, Fig. 4F and Fig 4e anywhere in the main manuscript.
- The processed data in Fig.4f looks noisy, particularly for ROIs 2 and 3.

This claim is really a stretch and should be removed.

We thank the reviewer for this suggestion. We have removed all claims related to “near cellular resolution” and added a new supplementary **Table S3** which includes the number of pixels and areas of each cluster in functional mouse brain imaging shown in Figure 4.

Minor Comments

The new Figure 2 as inserted in the manuscript pdf is pixelated.

We thank the reviewer for bringing this up. We believe this is due to the compression made by Microsoft Word. We will ensure the availability of all figures in high quality .pdf and .png formats.

A number paragraphs in the results section of the paper e.g., lines 199-214, lines 236-245 should really be in the discussion section?

We thank the reviewer for this suggestion, and we have moved the paragraph in lines 199-214 to the discussion, but prefer to keep the paragraph in lines 236-245 in their current location because they relate specifically to the microvascular imaging in the preceding section.

Reviewer #2:

The revised article “In vivo imaging with a flat, lensless microscope” by Adams et al. shows substantial improvement over the previous version. I thank the authors for refocusing the emphasis of the article on the live fluorescent labeled tissues and human mucosa. The additional experiment of the suspended fluorescent particles to demonstrate the 3D resolution in clear and scattering medium was also a wonderful addition and helped to prove the power and applicability of the technique, especially to scattering media like live tissue. Improvements to the text include moving some of the information and examples to the supplement, adding the additional fluorescent bead resolution demonstration and simulation, and clarifying many parts of the methods section and supplement. I especially appreciated the additional table information comparing BioFlatScope to other comparable techniques and technologies. One aspect of the BioFlatScope that I believe is most impactful and applicable is the high axial resolution demonstrated in scattering media compared to other techniques combined with a large FOV. I have no additional major concerns about the manuscript and believe it is appropriate for publication in Nature Biomedical Engineering.

However, I would ask that the authors add a few additional details to the text to help clarify some points or add needed information:

1) In the response to reviewers on page 20, you state “As Bio-FlatScope reported in this manuscript was designed to work at a relatively narrow working distance range (~1 mm)...” but then later state “The working distance ... for this specific Bio-FlatScope prototype is ~4mm.” This was confusing to me, can you please clarify? Is the working distance ~ 1mm or ~4mm? This does not need to be addressed in the manuscript text, just in the review response.

We apologize for this confusion caused by our wording and hope the following explanation helps to clarify: For each Bio-FlatScope prototype we operate over a 1 mm range of working distances (from 3.5 mm to 4.5 mm). We have updated that sentence to read:

“As Bio-FlatScope reported in this manuscript was designed to work at a relatively narrow working distance range of approximately 1 mm (from 3.5 mm to 4.5 mm), we did not characterize its lateral resolution in a large depth range. The lateral resolution acquired by analysing USAF target reconstructions remained consistent within the 3.5 mm to 4.5 mm depth range.”

2) The author's provided a nice description of the computational needs and performance of the algorithm on page 21 of the reviewer response, however it is not included in the methods section of the main manuscript. I would request that it be included in the manuscript or supplement.

We thank the reviewer for this suggestion and included the computational need of the algorithm in *Methods-Reconstruction*:

Lines 363-366:

"The current run time on an 8-Core Processor (AMD Ryzen 7 3700X, 3.59GHz) is around 1.3 s for a 2D frame containing 3072*2048 pixels using MATLAB. Each frame is stored as a 16-bit TIFF file. This runtime can likely be improved through algorithm optimization.”

3) While I appreciate the inclusion of the description about the phase mask in the supplement section 1 (lines 709-719), I was still left for more details about the fabrication of the phase mask. What resist was used, how was etching performed, what depths of etching or etching parameters (temperature, etc)? A basic recipe or additional details about the phase mask manufacturing should be included or referenced.

We thank the reviewer for this suggestion and enriched our description of phase mask fabrication in *Methods-Fabrication*:

Lines 315-323:

“The phase mask was fabricated using a 3D maskless two-photon photolithography system (Nanoscribe GmbH, Photonic Professional GT), with the High Resolution Dip-in Liquid Lithography (DiLL) configuration, using a photoresist (Nanoscribe, IP-Dip) on a 700 μm thick fused silica substrate. The laser power and writing speed may vary across systems and should be adjusted such that the pattern can be clearly observed in the real-time monitoring software (Nanowrite or AxioVision, with proper camera parameters adjustment). After the exposure, the sample was soaked in SU-8 developer for 20 minutes, and then soaked in isopropyl alcohol for 2 minutes. The fabricated phase mask has an overall height of 1 μm with a 200 nm height step, and a pixel size of 1 or 2 μm . “

Reviewer #4:

This paper proposes and prototypes a compact lens-less imaging device for in vivo fluorescence imaging, and conducts actual in vivo fluorescence imaging using the prototype device. The authors clarify the basic characteristics of the proposed device and show that it is useful for in vivo fluorescence imaging. The concept is different from that of a head-mounted miniature fluorescence microscopy system, and this reviewer judges that this is a significant paper that demonstrates a new application of lens-less imaging.

However, under constrained conditions, the in vivo fluorescence imaging system can be realized with existing microscope systems, and the reviewer finds it difficult to argue the superiority of the proposed method. The reviewer requests the authors to indicate whether the proposed method can be applied to free-living rodents and, if so, please describe the reason.

We thank the reviewer for bringing up the miniaturization of the imaging device, which is an important aspect in neural imaging device development. The proposed method can be applied to free-living rodents by miniaturizing the PCB supporting the image sensor as has been reported for other miniature head-mounted microscopes. The Bio-Flatscope prototype reported in this manuscript uses an off-the-shelf CMOS sensor and the total weight is approximately 5g. With additional miniaturization of the electronic components around the sensor, the weight of the entire device can be brought below 3g, which is suitable to be mounted on the head of freely-moving rodents.

Fig. S19. Respective weights of the components of Bio-FlatScope. Note that the off-the-shelf CMOS sensor contributes to the majority of the total weight, suggesting that Bio-FlatScope can be applied to free-living rodents by miniaturizing the PCB supporting the image sensor as has been reported for other miniature head-mounted microscopes.

We added this important future direction to the manuscript in *Discussion*:

Lines 288-296:

“One important opportunity for this class of lensless in vivo imagers is the potential to perform large area, high-resolution imaging in freely moving animals. While our current prototype, which weighs ~5 g, is too heavy for a freely moving mouse, nearly all of this

weight comes from the printed circuit board and electronic components that support the CMOS image sensor (Fig. S19). For other miniature head-mounted microscopes the sensor PCB and electronics have already been made light enough to be head mounted to a freely moving mouse (Table S1). Thus, using a similar design we expect future Bio-FlatScope will be compatible with freely moving mouse experiments.”

In addition, there are some points to be considered in this method, such as the following. The reviewer requests the authors to respond to these points.

(1) Excitation light source: The excitation light source layout is shown in Fig. S3, but the excitation light irradiation range seems to be narrow in this layout. Is there any way to widen the excitation range?

We thank the reviewer for bringing up the excitation light range which is important to the effective field of view of our device in *in vivo* imaging. The excitation light irradiation range of our fiber-optic cable and microprism can be increased by:

1. Increasing the NA of the commercially available fiber-optic cable. Please note that the increasing potential could be limited by the maximum allowable NA of optical fibers.
2. Increasing the distance between the microprism and the biological sample being illuminated.

For the specific need of our Bio-FlatScope prototype, the optimal excitation adjustment would be having a range equals to the FOV of the prototype, in the meantime maintaining sufficient illumination intensity, which could be compensated by tuning up the high-power excitation LED light source if larger excitation area is required.

We expanded our description of excitation light source in *Methods-Excitation Light*:

Lines 396-401:

“The excitation light irradiation area of our fiber-optic cable and microprism can be increased by: 1.) Increasing the NA of the fiber-optic cable (limited by the maximum NA of commercially available optical fibers), and 2.) Increasing the distance between the microprism and the biological sample being illuminated. For the specific needs of our Bio-FlatScope prototype, the optimal excitation adjustment would have an area equal to the FOV of the prototype while maintaining sufficient illumination intensity.”

(2) In this paper, the authors used a commercially available CMOS image sensor, which has a glass cover, so there is a certain distance between the phase mask and the imaging surface. Does this distance affect the spatial resolution of lens-less imaging?

We thank the reviewer for noting this technical detail, which we found essential to be considered during our prototyping. The lateral resolution of Bio-FlatScope can be approximately calculated by the following formula: (sensor pixel pitch) * (mask-to-scene distance) / (sensor-to-mask distance). Therefore, given a certain sensor-to-mask distance, high lateral resolution can be maintained by working at a proper mask-to-scene distance. We have added the following discussion to the manuscript in *Methods-Fabrication*:

Lines 309-314:

“The glass cover on the CMOS image sensor (~1.5 mm thick, including air gap) prevents ultra-close sensor-to-scene distances. In our case we operated with a ~4 mm working distance and a ~3.5 mm sensor-to-mask distance, so we did not need to remove this cover when building our prototypes. For closer sensor-to-mask prototypes one may choose sensors with removable glass cover (e.g. Rasp Pi Camera), or with a smaller gap between the glass cover and the active sensor surface.”

(3) It is difficult to prevent autofluorescence from the absorption filter by using only two types of filters, absorption filter and interference filter (refer to the following paper

K. Sasagawa et al., "Highly sensitive lens-free fluorescence imaging device enabled by a complementary combination of interference and absorption filters," *Biomedical optics express*, vol.9, no.9, pp.4329-4344, 2018.)

Please comment on this point.

We thank the reviewer for referring to this work, in which the authors report a triple layer hybrid filter, demonstrate its excellent rejection ratio of $10^8:1$, and show that it sufficiently supports fluorescence imaging under a contact imaging situation. One important difference between the imaging configuration in our manuscript compared to this reference is the longer working distance we use for Bio-FlatScope, which helps reduce the effects of autofluorescence. We have added the following description to the manuscript in *Method-Fabrication*:

Lines 328-337:

“One potential challenge for lensless fluorescence imaging is the fact that absorption filters designed to filter the excitation light can themselves produce autofluorescence. One approach to overcome this issue was demonstrated by Sasagawa et al., which used a fiber optic plate to help collimate the autofluorescence so that it would be better rejected by an interference filter [70]. In our case, however, we found that increasing the working distance and separation between absorption filter and sensor for FlatScope [10] and Bio-FlatScope allowed us to achieve sufficient fluorescence contrast because it reduced the amount of excitation light incident on the filter, and improved the performance of the interference filter by constraining the angle of the incident light.”